# Spatiotemporal proteomics reveals the biosynthetic lysosomal membrane protein interactome in neurons

Chun Hei Li [1], Noortje Kersten [1], Nazmiye Özkan [1], Dan T. M. Nguyen [1], Max Koppers [1,2], Harm Post [3], Maarten Altelaar [3] & Ginny G. Farias [1] ✉

Lysosomes are membrane-bound organelles critical for maintaining cellular homeostasis. Delivery of biosynthetic lysosomal proteins to lysosomes is crucial to orchestrate proper lysosomal function. However, it remains unknown how the delivery of biosynthetic lysosomal proteins to lysosomes is ensured in neurons, which are highly polarized cells. Here, we developed Protein Origin, Trafficking And Targeting to Organelle Mapping (POTATO-Map), by combining trafficking synchronization and proximity-labelling based proteomics, to unravel the trafficking routes and interactome of the biosynthetic lysosomal membrane protein LAMP1 at specified time points. This approach, combined with advanced microscopy, enables us to identify the neuronal domain-specific trafficking machineries of biosynthetic LAMP1. We reveal a role in replenishing axonal lysosomes, in delivery of newly synthesized axonal synaptic proteins, and interactions with RNA granules to facilitate hitchhiking in the axon. POTATOMap offers a robust approach to map out dynamic biosynthetic protein trafficking and interactome from their origin to destination.

Lysosomes play a central role in maintaining cellular homeostasis. They participate in numerous processes such as protein and organelle quality control and degradation, nutrient sensing, metabolic signaling and membrane repair, which are critical during development and ageing[1–3]. An outstanding question is how the lysosomal proteome is maintained throughout the lifespan of a cell, particularly in non-dividing cells such as neurons.

Lysosomal proteins are synthesized in the endoplasmic reticulum (ER), sorted to the Golgi and packaged for their delivery to lysosomes[4]. At the steady state, these proteins are already distributed in immature and mature lysosomes, making it difficult to study the biosynthetic pool, including mechanisms of trafficking, targeting, regulation and specific functions. Among lysosomal proteins, LAMP1 and LAMP2, proposed to be essential in lysosome maintenance, are the most abundant proteins lining the limiting membrane of a lysosome[5,6]. Their essential role has been demonstrated in mice, in which double knockout causes lethality[7].

In unpolarized cells, the sorting of biosynthetic LAMPs to lysosomes has been proposed to occur via direct targeting to immature or mature lysosomes, or via indirect targeting via fusion with the plasma membrane or early endosomes[2,8,9]. However, in highly polarized neurons, it remains unknown how biosynthetic LAMPs are sorted, transported and targeted to lysosomes. In these cells, a logistic challenge arises for delivery of biosynthetic LAMPs to lysosomes, as lysosomes are not restricted to the soma, but are also present in dendrites and along the extremely long axon[10,11]. Dysregulation of lysosome biogenesis and function have been linked to several neurological disorders, which highlights the need to better understand how lysosomes are maintained in neurons[12].

[1]Cell Biology, Neurobiology and Biophysics, Department of Biology, Faculty of Science, Utrecht University, Utrecht 3584 CH, The Netherlands. [2]Center for Neurogenomics and Cognitive Research, Department Functional Genomics, Vrije Universiteit Amsterdam, Amsterdam, Netherlands. [3]Biomolecular Mass Spectrometry and Proteomics, Bijvoet Center for Biomolecular Research and Utrecht Institute for Pharmaceutical Sciences, Utrecht University, Utrecht, Netherlands. ✉e-mail: g.c.fariasgaldames@uu.nl

Recent methods such as the retention using selective hooks (RUSH) system and derived tools, have enabled visualization of the sorting of biosynthetic proteins to their final destinations by synchronizing protein release from the ER[13,14]. However, these methods require co-labeling with different markers to at least pinpoint their location at a specific time point, and do not enable characterization of the full composition of compartments carrying biosynthetic proteins.

Here, we developed an approach that can map the interactome of biosynthetic proteins with high spatiotemporal precision in neurons, which we termed Protein Origin, Trafficking And Targeting to Organelle Mapping (POTATOMap). POTATOMap tracks the route of biosynthetic lysosomal proteins from the moment they exit the ER to their traversal through the biosynthetic and endolysosomal pathways, generating snapshots of the organelle proteome and interactome at specified time points.

We show that our whole-neuron POTATOMap approach can systematically identify machineries associated with biosynthetic LAMP trafficking and sorting at the ER, Golgi, post-Golgi, endosomes and lysosomes. Interestingly, we reveal neuronal domain-specific mechanisms for biosynthetic LAMP targeting into the axon, including motors, adaptors and SNAREs involved in trafficking and fusion of biosynthetic LAMPs. Moreover, we identify three different roles of biosynthetic LAMP trafficking into the axon: axonal lysosome replenishment, delivery of axonal synaptic proteins, and contact and hitchhiking of RNA on biosynthetic LAMP compartments. Together, our findings suggest important yet under-explored roles of biosynthetic LAMP compartments prior to reaching axonal lysosomes.

POTATOMap offers the opportunity to address questions requiring precise spatiotemporal resolution of membrane protein trafficking routes and identification of transient interactions with potential key regulatory players, in health and disease.

## Results

### Delivery of biosynthetic LAMP proteins into lysosomes in neurons

To develop a method to track biosynthetic LAMPs and their associated interactome over time, we first aimed to determine their dynamics in polarized neurons. Thus, we used the RUSH system, which allows the visualization of biosynthetic protein synchronized release from the ER and sorting along the biosynthetic pathway. This works through ER retention by heterodimerization of a Streptavidin (Strep) fused to a retention signal or hook (KDEL) and Streptavidin Binding Peptide (SBP) fused to a protein of interest. Upon addition of biotin, which competes for binding to Strep, the protein is released from the hook, and trafficking is synchronized. The RUSH system was generated for the lysosomal transmembrane proteins LAMP2A and LAMP1, two of the most abundant and commonly used lysosome markers (Fig. 1a). Biosynthetic lysosomal protein distribution and dynamics were analyzed in polarized rat hippocampal neurons on day-in-vitro (DIV) 6–8. mNeonGreen(mNG)-tagged RUSH-LAMP2A and mScarlet-I-tagged RUSH-LAMP1 were properly retained along the ER prior to biotin addition, and they had kinetics for synchronized cargo release and trafficking from the ER to Golgi (20–30 min) similar to previous studies (Fig. 1b; Supplementary Fig. 1a)[15]. Within 1 h of biotin addition, RUSH-LAMP2A-positive compartments were clearly visualized (Fig. 1b).

To determine whether biosynthetic lysosomal proteins are delivered into pre-existing lysosomes after 1 h of release, we combined RUSH-LAMP2A-mNG with LAMP1-RFP, expressed for 16 h. To dissect between immature and mature lysosomes, neurons were live-labeled with SirLyso (cathepsinD-activity probe), during the last 20 min of cargo release. Many tubular structures positive for RUSH-LAMP2A were observed, but most of them were not incorporated into immature or mature lysosomes after 1 h of release (Fig. 1c–e). We then assessed whether, in neurons, biosynthetic LAMPs are delivered to lysosomes 4 h after release from the ER. We observed that most of the

RUSH-LAMP2A compartments were targeted to immature or mature lysosomes 4 h after release (Fig. 1f, g). Similar results were observed 24 h after release (Supplementary Fig. 1b, c).

These results indicate that in neurons biosynthetic LAMPs are sorted from the Golgi into compartment(s) distinct from pre-existing lysosomes at 1 h, and that they have reached lysosomes at 4 h.

### Spatiotemporal interactome of biosynthetic LAMP in neurons

Because of the unique nature of biosynthetic LAMP compartments observed at 1 h, we wondered whether the composition and interactome of these compartments are distinct from pre-existing lysosomes at 4 h. We therefore developed a system to map the temporal interactome of biosynthetic compartments, which we termed Protein Origin, Trafficking And Targeting to Organelles Mapping, the POTATOMap system.

In this system, we tagged biosynthetic LAMP1 with APEX2, a peroxidase used for proximity labeling-based proteomics to elucidate its organelle proteome and interactome at different time points (Fig. 2a, b)[16]. These time points were determined based on our imaging data (Fig. 1). We first characterized the localization of RUSH-LAMP1-V5-APEX2 and proximal biotinylated endogenous proteins, after cargo release from the ER, by adding biotin and/or biotin-phenol for 20 min, 1 hour and 4 hours, with and without $H_2O_2$ (Fig. 2a–d). This approach allowed for a robust synchronization of biosynthetic LAMP trafficking to different compartments and labeling of endogenous proteins in close proximity over time (Fig. 2c, d). To obtain a comprehensive temporal interactome of biosynthetic LAMP compartments, we performed proximity labeling and protein isolation at designated time points, followed by streptavidin-pulldown and LC-MS/MS (Fig. 2b).

POTATOMap identified 580 proteins enriched at 3 different time points (65 proteins at 20 min, 387 proteins at 1 h and 276 proteins at 4 h) (Fig. 2e). Comparative analysis of the 3 datasets revealed a remarkably clear segregation of protein and pathway enrichment between time points (Fig. 2e; Supplementary Fig. 2a–c). Gene ontology (GO) analysis of significantly enriched proteins at 20 min showed an abundance of processes involved in the early secretory pathway, including processes essential for biosynthetic LAMP secretion through the ER−ERGIC−cis-Golgi compartments (Fig. 2f). Processes specifically related to trans-Golgi network (TGN), post-Golgi transport, endosome transport were enriched at 1 h, indicating the trafficking of biosynthetic LAMP in the late secretory pathway and crosstalk with the endocytic and recycling pathways (Fig. 2g). Interestingly, processes related to synaptic components, RNA metabolism and neurite projections were the most enriched at 1 h, suggesting a remarkably intertwined connection between biosynthetic LAMP trafficking, neuron development and synaptic function (Fig. 2g). Additional cellular processes related to endosomal fusion machinery and axonal maintenance were shared between 1 h and 4 h (Supplementary Fig. 2d–g). On the other hand, cellular processes associated with lysosomal function, mTOR signaling and amino acid sensing machineries were highly enriched at 4 h (Fig. 2h).

Our comparative POTATOMap datasets show a clear shift in organelle identity and interactome. We also reveal processes known to be associated mainly with lysosomes (4 h), are also present within biosynthetic LAMP compartments (1 h).

### POTATOMap reveals distinct proximity proteomes for biosynthetic LAMP over time

We further investigated the proteins according to their related biological processes and highlighted several key proteins with their respective enrichment at specific time points. In the early secretory pathway, biosynthetic LAMPs traffic from the ER to the cis-Golgi. Consistent with this, we found that proteins related to ER Exit Sites (ERES) including SEC23A, three homologs of SEC24A, SEC24B and SEC24C were enriched at 20 min (Fig. 3a). Subsequently, biosynthetic

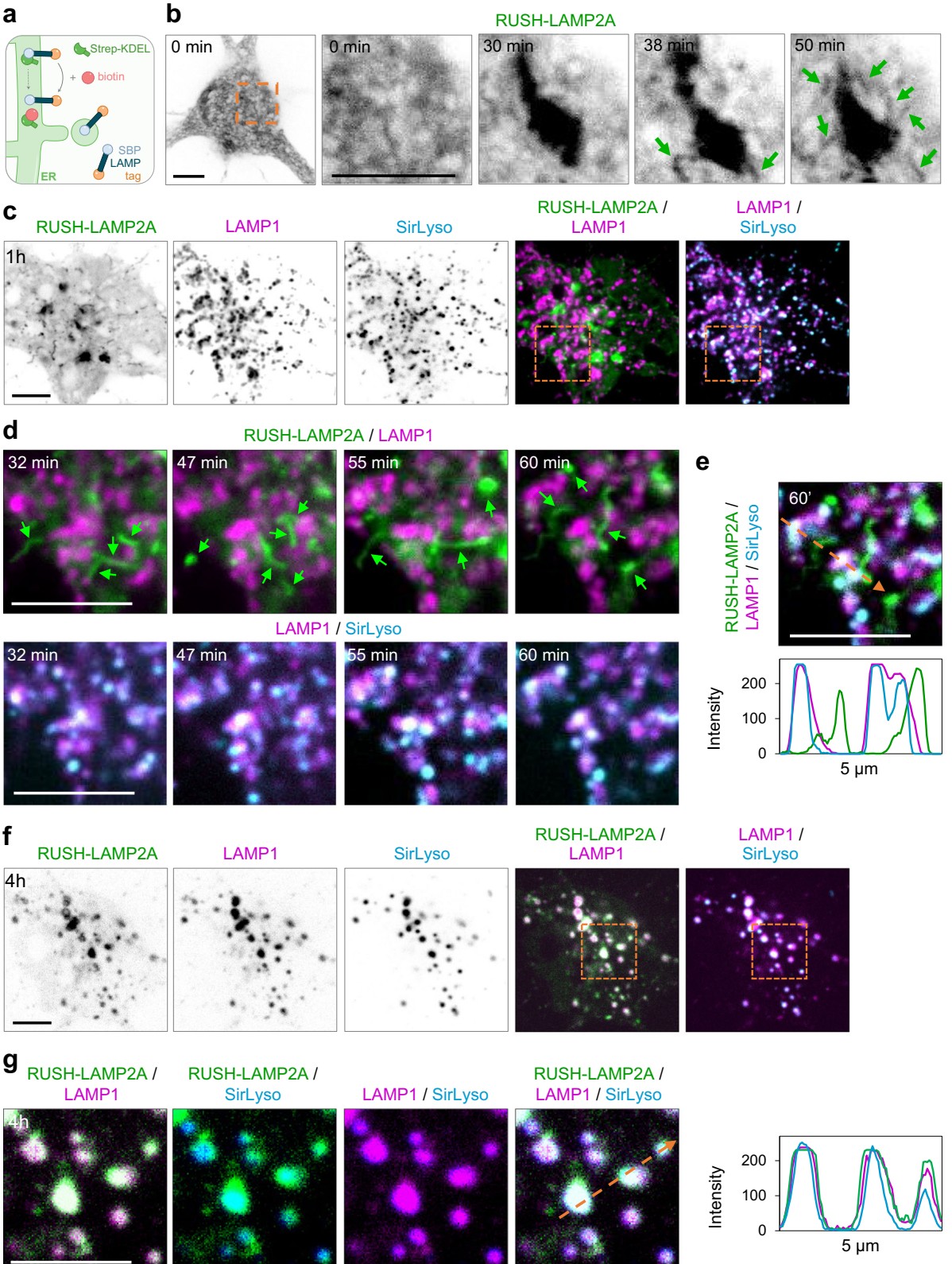

**Fig. 1 | Biosynthetic LAMP is present in an organelle distinct from lysosomes at 1 h after ER exit, but it fuses with lysosomes at 4 h. a** Schematic showing the RUSH system for retention and release of biosynthetic LAMPs. **b** Representative still images from live hippocampal neurons expressing RUSH-LAMP2A-mNG and imaged on a confocal spinning disk immediately after biotin addition every 1 min for 1 h. **c–g** Representative still images from neurons co-expressing RUSH-LAMP2A-mNG and LAMP1-RFP (to visualize its steady state lysosomal distribution) and live-labeled with SirLyso 20 min prior to imaging to visualize mature lysosomes. Time after biotin addition is indicated in images. Selected magnified region in orange boxes. Orange arrows indicate the region used to generate intensity profile graphs. Green arrows point to biosynthetic LAMP2A-positive tubular organelles. Representative images reported in (**b–g**) were repeated in at least three independent experiments. Scare bar, 5 μm. See also Supplementary Fig. 1.

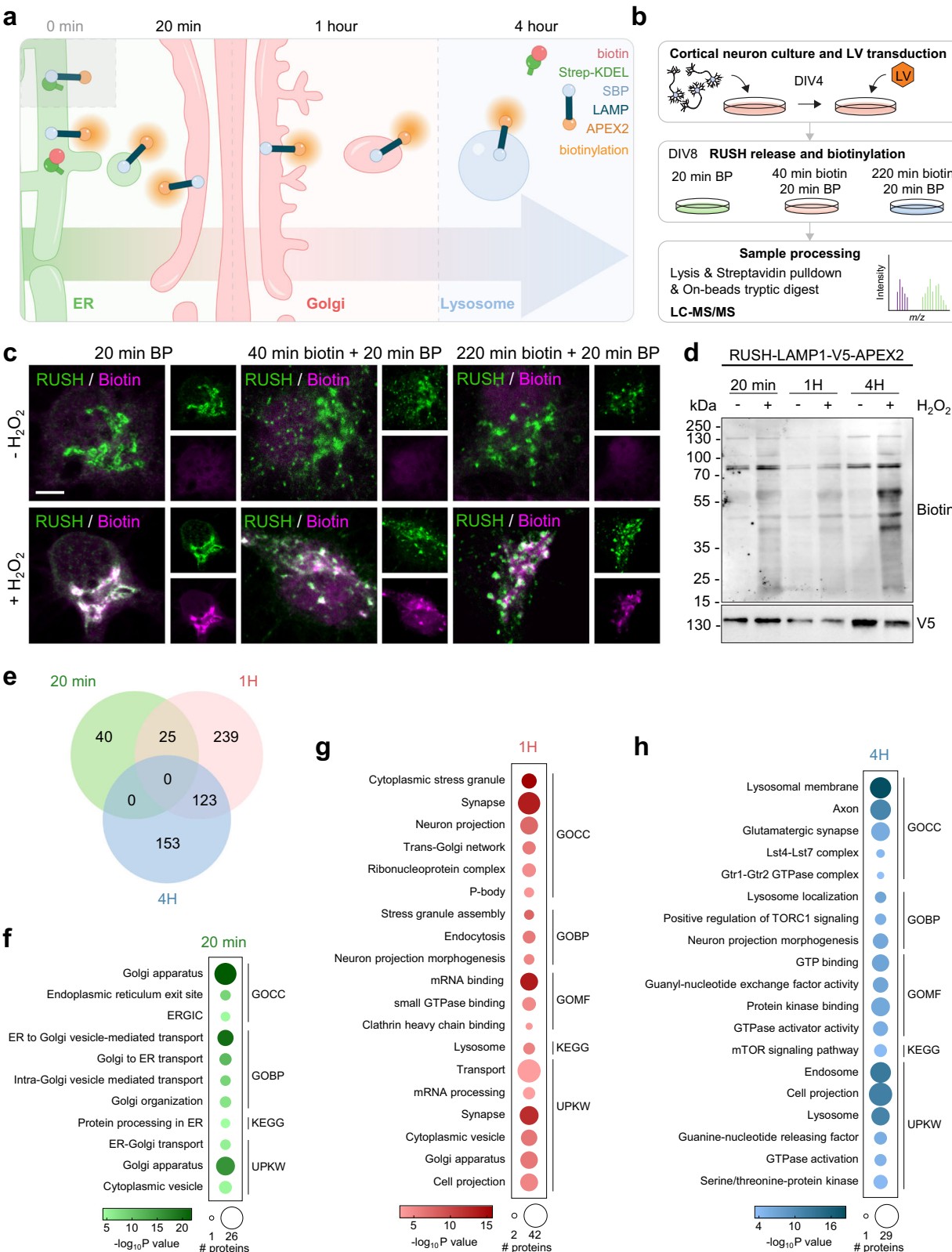

LAMPs reach the TGN and exit in post-Golgi compartment, which is reflected by enrichment of ARF1, ARF5, AP1, AP3, AP4, RAB6A and RAB6B (Fig. 3a)[17–19]. This comparative dataset also identified the presence of Golgi-endosome tethering complex components, such as FAM91A1, TBC1D23 and WDR11 at both 20 min and 1 h (Fig. 3a)[20]. After biosynthetic LAMPs exit the TGN, proteins related to endocytic recycling such as RAB11B, RAB11FIP5, RAB4B, the EARP complex VPS50,

VPS51, VPS52 and VPS53 were enriched at 1 h (Fig. 3a)[21–23]. Collectively, these enriched proteins involved in early secretion, post-Golgi exit and trafficking, were significantly reduced as the biosynthetic lysosomal compartment matured (Fig. 3a). Notably, proteins enriched at 4 h were mostly endo-lysosomal proteins, previously identified and characterized by various proteomics approaches such as endo/lyso-IP or proximity labeling of lysosomes with APEX2, at endogenous and

**Fig. 2 | POTATOMap dissects the biosynthetic LAMP-positive compartment interactome over time. a** Illustration of POTATOMap (Protein Origin, Trafficking And Targeting to Organelle Mapping) paradigm for biosynthetic LAMP. **b** Conditions and workflow for mass spectrometry in cortical neurons. **c** Representative confocal images of neurons expressing RUSH-LAMP1-V5-APEX2 treated with biotin-phenol (BP) only or biotin and biotin-phenol as indicated. Control neurons without $H_2O_2$ (top panel) and biotinylated neurons with $H_2O_2$ (bottom panel) were immunostained with antibodies against V5 (RUSH) and 555-conjugated Strep (biotin). Scale bar, 5 μm. **d** Immunoblot of RUSH-LAMP1-V5-APEX2 transduced neurons lysed at three time points as indicated, with and without $H_2O_2$. **e** Venn diagram of proteins significantly enriched at their respective time point(s) (*N* = 2 independent experiments with 2 technical repeats per experiment, two-sided

t-test, *p* value < 0.05, log$_2$ fold change ≥ or ≤±1, no multiple comparison test was used). **f–h** DAVID analysis of Gene Ontology (GO) terms enriched among the proteins specific for each time point in (e, default setting was applied to DAVID analysis to determine process enrichments. Processes were first filtered by FDR < 0.01; then by EASE score, modified Fisher Exact *p* value < 0.05). Dot size represents the number of different proteins, dot color represents respective time point and term enrichment (*p* value < 0.05). GOCC, cellular compartment; GOBP, biological process; GOMF, molecular function; UPKW, Uniprot keyword. Representative image in **c**, **d** were repeated in at least 3 and 2 independent experiments respectively. See also Supplementary Figs. 2 and 3 and Source Data 1. Source data are provided as a Source Data file.

---

exogenous levels, in neuronal and non-neuronal cells (Supplementary Fig. 3a–d)[24–32]. Specifically, proteins known to be a crucial part of the nutrient-sensing mTOR signaling pathway such as LAMTOR1/2/3, RRAGA/C and GATOR2 protein complexes were significantly enriched at 4 h (Fig. 3a)[33,34]. POTATOMap also identified proteins involved in endo-lysosomal trafficking such as ARL8A, ALR8B, KIF1A, KLC2, PIP4P1 and PIP4P2[35–38]. These proteins displayed moderate enrichments at 1 h but were significantly enriched at 4 h compared to baseline (20 min), indicating a stable and continuous involvement of these trafficking machineries over time (Fig. 3a).

We further validated our comparative proteome analysis with confocal imaging. We studied the co-distribution of RUSH-LAMP1 with proteins enriched after 1 h and/or 4 h of biotin addition. The SNARE STX6 was co-distributed with LAMP1 at the Golgi, while they distributed in distinct compartments at 4 h (Fig. 3b; Supplementary Fig. 4a). RAB11-positive recycling endosomes were in close proximity to LAMP compartments at 1 h, but this proximity was reduced at 4 h (Fig. 3c; Supplementary Fig. 4b). In contrast, RAB7-positive late endosomes were only in close proximity to biosynthetic LAMP1 compartments at 1 h, but they co-distributed at 4 h (Fig. 3d; Supplementary Fig. 4c). Lastly, the mTOR signaling protein LAMTOR4 co-distributed with LAMP1 compartments at 4 h but not at 1 h (Fig. 3e; Supplementary Fig. 4d).

Thus, our POTATOMap approach identifies previously reported key players associated with LAMP1 origin, trafficking and targeting to lysosomes with high spatiotemporal resolution.

### Biosynthetic LAMP is delivered to the axon, in a compartment distinct from lysosomes, to replenish axonal lysosomes

Our proteomics data revealed an enrichment of processes associated with the axonal domain at 1 h and 4 h (Fig. 2g, h; Fig. 3a; Supplementary Fig. 2e). We therefore wondered if biosynthetic LAMP compartments can be trafficked into the axon. We quantified their number and tracked their transport direction in a proximal part of the axon immediately after the axon initial segment (AIS). Biosynthetic LAMP compartments were long-range transported along the axon, in which most of them underwent anterograde transport into the axon 1 h after release (Fig. 4a, b).

Our whole neuron POTATOMap data showed that biosynthetic LAMP compartments between 1 h and 4 h post-release are distinct in nature, with most known lysosomal processes found mainly at 4 h, and not at 1 h (Fig. 2a). We evaluated whether this is also the case for the axon-specific LAMP compartment population. Along the axon, antero- and retrograde transport of immature and mature lysosomes was observed. However, most biosynthetic LAMP compartments did not co-traffic with pre-existing lysosomes (Fig. 4c, d, g, h; Supplementary Fig. 5a–f). Also, the transport velocity of biosynthetic LAMP compartments at 1 h was higher compared to lysosomes moving anterogradely at 4 h (Supplementary Fig. 5g). Cathepsin activity was not observed in biosynthetic compartments negative for pre-existing LAMP (Fig. 4c, d, g, h; Supplementary Fig. 5a–f). These results indicate that biosynthetic

LAMP proteins are predominantly delivered into the axon in a compartment distinct from pre-existing lysosomes.

We wondered whether these biosynthetic LAMP compartments entering the axon mature themselves, or whether these are delivered to the axon for replenishment of pre-existing lysosomes. To assess this, we analyzed the dynamics of RUSH-LAMP2A 2 h and 4 h after release in neurons labeled for pre-existing immature and mature lysosomes with LAMP1 and SirLyso. Two hours after RUSH-LAMP2A release, biosynthetic LAMP compartments overlapped with stationary and retrogradely transported lysosomes, while only a few of antero-grade transported RUSH-LAMP2A co-localized with pre-existing lysosomes (Fig. 4e, g, h; Supplementary Fig. 5a–f). Four hours after release, many RUSH-LAMP2A compartments were present within stationary, retrograde and anterograde moving lysosomes (Fig. 4f, g, h; Supplementary Fig. 5a–f). Importantly, LAMP compartments negative for pre-existing LAMP1 were mostly negative for SirLyso, after 2 h and 4 h release (Fig. 4e–h; Supplementary Fig. 5a–f).

These results show that our whole-neuron POTATOMap approach can also represent the domain-specific lysosomal protein interactome. Our findings also suggest that biosynthetic LAMPs targeted into the axon play an important role in axonal lysosome replenishment.

### RAB6A-decorated biosynthetic LAMP compartments require kinesins 1 and 3, and the adaptor protein ARL8B for transport to the axon.

In our POTATOMap proteomics datasets, we found that out of 45 kinesins, 8 kinesins were associated to biosynthetic LAMP compartments at 1 h and/or lysosomes at 4 h (Fig. 5a). Two major kinesin motors transport cargoes into the axon, kinesin 1 (heavy chain KIF5A-C and light chain KLC1-2), and kinesin 3 (KIF1A and 1Bβ). For instance, kinesin 1 transports lysosomes, mitochondria and ER into the axon, while kinesin 3 transports axonal synaptic vesicles[10,39–41]. We found that the transport of biosynthetic LAMP compartments 1 h after release was disrupted after knocking down KIF5A-C or KIF1A, with a more severe phenotype for KIF1A knockdown (Fig. 5b, c; Supplementary Fig. 6a–c, e). Trafficking from the Golgi was not disrupted upon KIF5A-C or KIF1A knockdowns (Supplementary Fig. 6g).

A previous study found that ARL8B together with the BORC complex links lysosomes to kinesin 1 and kinesin 3 in cell lines[37], and kinesin 1 together with ARL8B and the BORC complex are involved in the transport of lysosomes into the axon[10]. Consistent with this, we found ARL8B and BORC subunits significantly enriched at 4 h. However, we also observed a significant increase of ARL8A-B between 20 min and 1 h (Fig. 5a). We wondered whether ARL8 was also required for the transport of biosynthetic LAMP compartments. ARL8B disruption caused reduced transport of RUSH-LAMP1 after 1 h of release, while trafficking from the Golgi was not disrupted (Fig. 5b, c; Supplementary Fig. 6d, g). The knockdown of the BORC subunit BORCS5 caused just a slight reduction in the axonal transport of RUSH-LAMP1 at 1 h, aligned with our proteomics data in which BLOC1S1 (BLOS1) and SNAPIN, two other BORC subunit were uniquely associated to lysosomes at 4 h (Supplementary Fig. 2h).

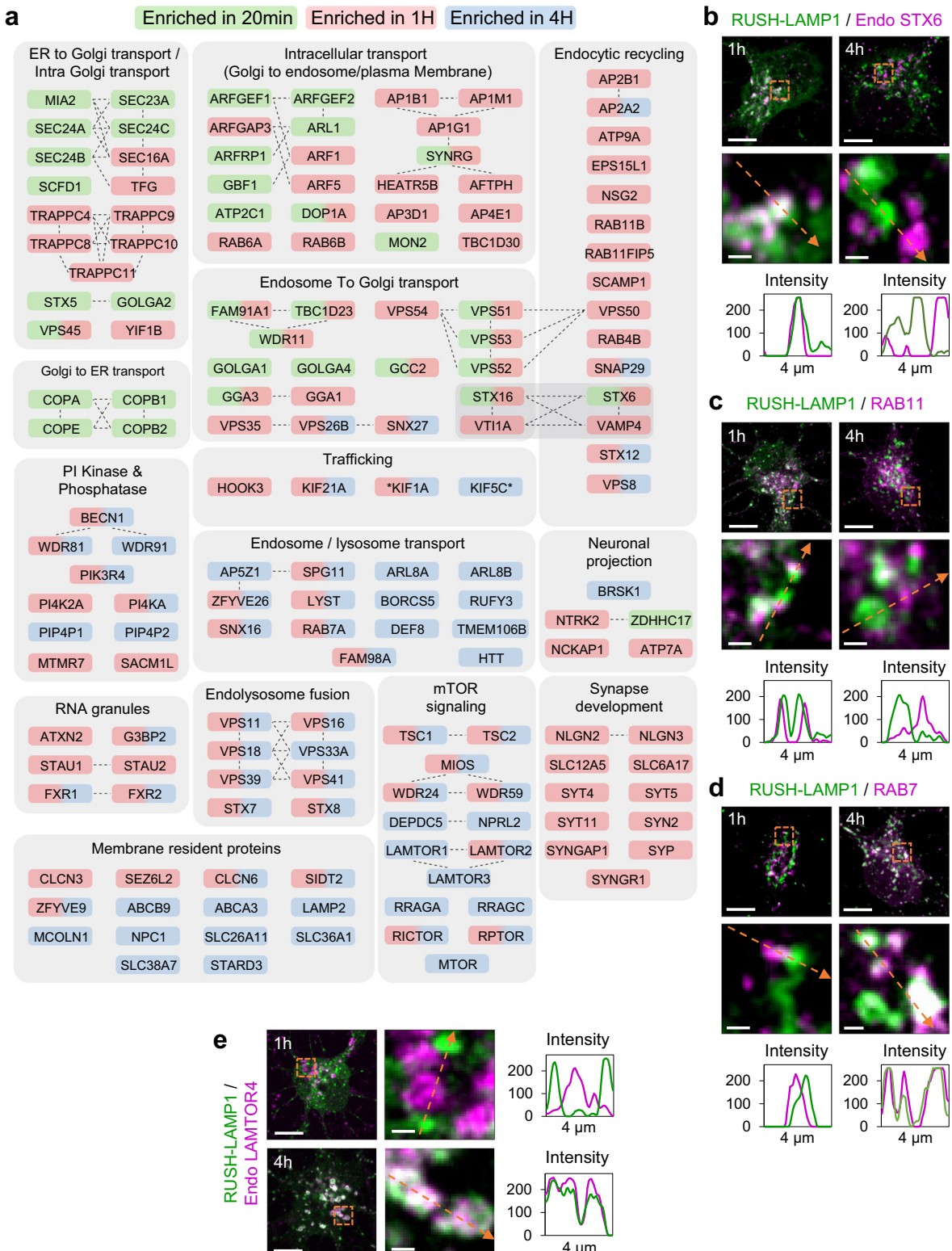

**Fig. 3 | POTATOMap reveals interactors and potential key proteins for biosynthetic LAMP compartment trafficking and targeting to lysosomes.**
**a** Schematic representation of identified proteins within enriched GO terms. Color coding indicates the time point(s) at which a given protein is enriched (Green 20 min; Red 1 hour; Blue 4 hour). Dashed lines between proteins specify physical interactions analyzed with STRING; grey rectangle depicts proteins involved in both processes. All indicated proteins are statistically significant ($p$ value < 0.05). Asterisks indicate that the Log$_2$ fold change enrichment at that time point is smaller

than 1. **b–e** Confocal images of neurons co-labeled for RUSH-LAMP1-V5 and STX6 (**b**), RAB11 (**c**), RAB7 (**d**), and LAMTOR4 (**e**) after 1 h or 4 h of biotin addition. Magnified regions from soma are depicted with orange boxes. Orange arrows indicate the region used to generate intensity profile graphs. Scale bar, 10 μm, and magnified images, 1 μm. Representative images in **b–e** were repeated in at least 2–3 independent experiments. See also Supplementary Figs. 2 and 4, and Source Data 1. Source data are provided as a Source Data file.

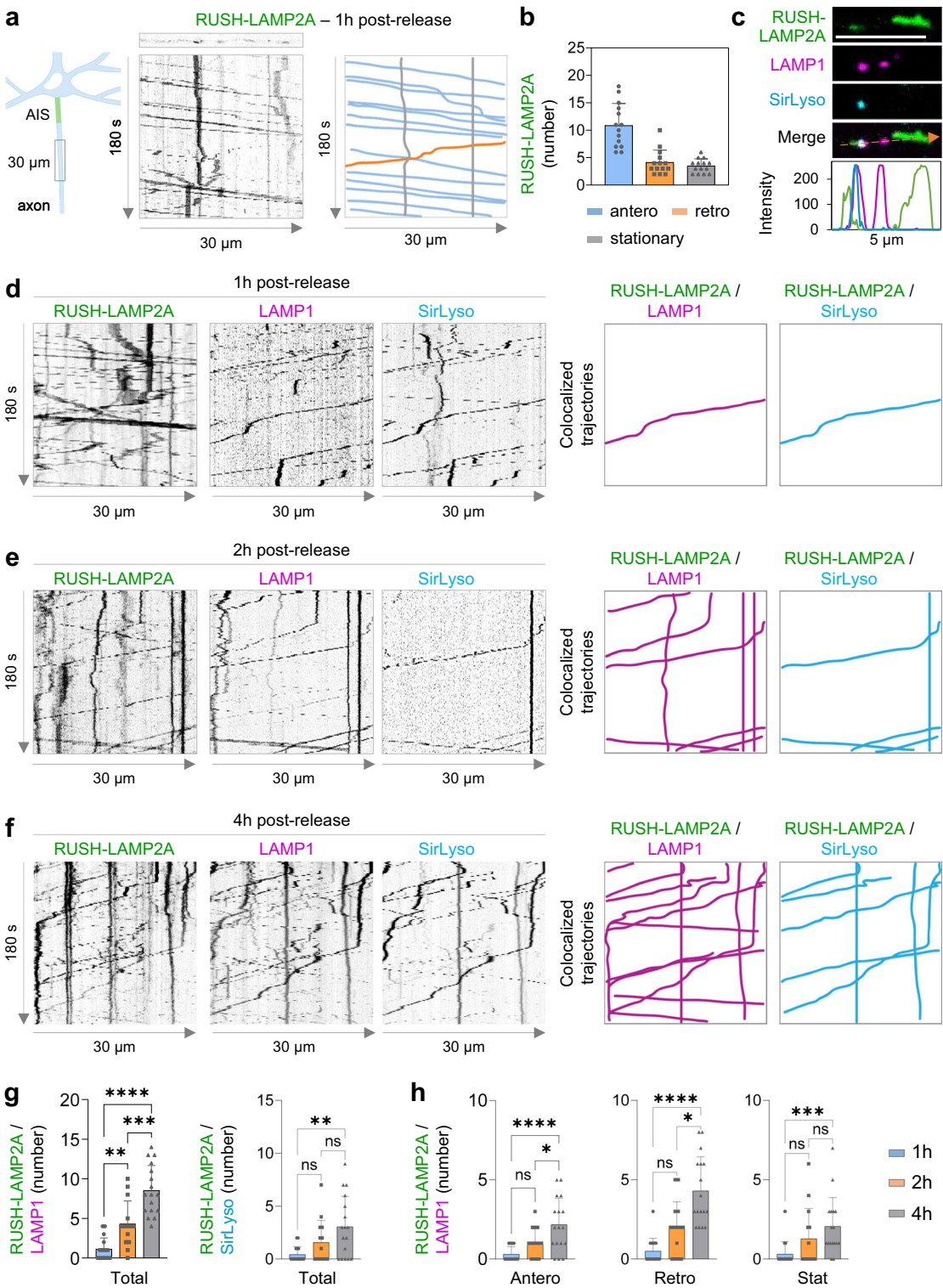

Although biosynthetic LAMP compartments share similar machinery with lysosomes for their transport into the axon, we wondered whether we could identify a protein that specifically labels the biosynthetic LAMP compartment prior to its fusion with lysosomes. In our proteomics dataset, RAB6 was particularly enriched at 1 h and decreased at 4 h (Fig. 5a). We studied the co-distribution of RAB6A and RUSH-LAMP1 1 h after release. RUSH-LAMP1 colocalized with RAB6A at the Golgi with tubules budding from the Golgi (Fig. 5d, e). We also observed a high overlap between RUSH-LAMP1 and RAB6A 1 h after release along the axon, which was drastically reduced after 4 h

(Fig. 5f, g; Supplementary Fig. 6i). We reasoned that we could use the overlap between RAB6A and LAMP1 to determine the percentage of LAMP compartments that are biosynthetic in nature at the steady state. We estimate that ~11% of all LAMP1-positive organelles correspond to biosynthetic LAMP compartments at the steady state (Fig. 5h–i; Supplementary Fig. 6j). Interestingly, we also observed RAB6A-positive compartments in close proximity to lysosomes, which suggest the transfer of biosynthetic LAMP proteins for replenishment (Fig. 5j).

Thus, biosynthetic LAMP compartments require both kinesin-1 and kinesin-3, as well as the motor adaptor protein ARL8. These

**Fig. 4 | Biosynthetic LAMP enters the axon as a distinct compartment other than lysosomes, but it fuses with and replenishes axonal lysosomes. a** Left, schematic showing the acquisition region of RUSH-LAMP2A-mNG transport along the axon. Right, kymographs generated from live cell imaging along the axon every 1 s for 180 s are shown. Anterograde, retrograde, and stationary direction is depicted with blue, orange, and grey lines. **b** Number of RUSH-LAMP2A-positive compartments moving antero- or retrogradely, or stationary after 1 h of biotin addition. $n = 14$ neurons, from 3 independent experiments ($N = 3$). **c** Still image from the axon of a live neuron expressing RUSH-LAMP2A-mNG and LAMP1-RFP, labeled with SirLyso. Orange arrow indicates region used to generate the intensity profile graph. Scale bar, 5 μm. **d–f** Kymographs of live neurons expressing RUSH-LAMP2A-mNG and LAMP1-RFP, labeled with SirLyso, after 1 h (**d**), 2 h (**e**) and 4 h (**f**) of biotin addition. Colocalized trajectories are traced and plotted on the right.

**g** Quantification of total number of colocalized trajectories after 1, 2 and 4 h of biotin addition between RUSH-LAMP2A and LAMP1 (**p = 0.0051, ****p < 0.0001, ***p = 0.0001) or RUSH-LAMP2A and SirLyso (ns p = 0.465, **p = 0.0068, ns p = 0.3961). $n = 16$, 14 and 17 neurons, respectively; each from 3 independent experiments ($N = 3$). **h** Quantification of total colocalized compartments between RUSH-LAMP2A and LAMP1 in anterograde (ns p = 0.1836, ****p < 0.0001, *p = 0.0296), retrograde (ns p = 0.0944, ****p < 0.0001, *p = 0.0193) and stationary (ns p = 0.336, ***p = 0.0007, ns p = 0.1555). Data are presented as mean values ± SD, plus individual points. Ordinary one-way ANOVA test followed by a Tukey's multiple comparison test for left graph in (**g**); middle graph in (**h**). Kruskal–Wallis test followed by a Dunn's multiple comparison test for the right graph in (**g**) and the middle and right graphs in (**h**). See also Supplementary Fig. 5. Source data are provided as a Source Data file.

---

compartments are RAB6A-decorated, indicating they are post-Golgi carriers in nature, and they represent ±11% of LAMP-positive organelles along the axon, at the steady state.

## Biosynthetic LAMPs and axonal presynaptic proteins are co-trafficked into the axon prior to their segregation into distinct compartments

Intriguingly, we found axonal pre-synaptic proteins in our proteomics data for RUSH-LAMP1 after 1 h release (Fig. 6a). These include SYN1, SYN2, SYNGAP1, SYNGR1, SYP, and SYT1, which were enriched at 1 h, compared to 20 min and 4 h (Fig. 6a).

The presence of synaptic proteins in our POTATOMap dataset with temporal specificity motivated us to explore the possible co-trafficking route of newly synthesized synaptic and LAMP proteins. We performed live cell imaging with simultaneous RUSH for both LAMP2A and the axonal synaptic protein SYT1 after 1 h or 4 h of release. In the soma, we observed compartments containing both proteins and budding from the Golgi at 1 h (Fig. 6b). To ensure that the co-budding of these cargoes from the Golgi was not due to Golgi overload, we tested the secretion of the hydrolase CTSB. We observed that RUSH-LAMP2A and RUSH-CTSB left the Golgi in morphologically distinct compartments, as previously reported in cell lines (Supplementary Fig. 7a)[15,42].

Along the axon, we observed abundant anterograde co-transport of both LAMP2A and SYT1 proteins, consistent with the biosynthetic nature of LAMP compartments at 1 h (Fig. 6c; Supplementary Fig. 7b). This co-transport was significantly reduced after 4 h, suggesting they segregate prior to the fusion of biosynthetic LAMPs with pre-existing axonal lysosomes (Fig. 5h–j; Fig. 6c, d; Supplementary Fig. 7b).

Our results show that biosynthetic LAMP and synaptic vesicle proteins are sorted together from the Golgi into post-Golgi compartments, which are co-transported into the axon to possibly replenish not only lysosomes, but also synaptic vesicles.

## The SNARE VAMP4 is required for the transport of LAMPs and synaptic proteins into the axon

Our proteomics data also revealed the association of membrane fusion machineries, including specific SNARE proteins at specific time points (Fig. 6a). Some of these SNARE proteins have been reported to regulate lysosome maturation in cell lines, and synaptic vesicle function. An interesting SNARE candidate is VAMP4, which has been associated with the targeting of insulin-containing secretory granules to lysosomes in insulinoma cells, and in the recycling of synaptic vesicles at axon terminals in neurons[43,44]. However, it is unknown whether VAMP4 plays a role in the delivery of biosynthetic synaptic and LAMP proteins to the axon. In neurons, we found VAMP4 was enriched in the Golgi, and colocalized with RUSH-LAMP1 at 1 h in the Golgi and to a lesser extent also with biosynthetic LAMP compartments budding from the Golgi (Fig. 6e). We found that VAMP4 knockdown reduced the transport of both RUSH-LAMP1 and RUSH-SYT1 along the axon after 1 h (Fig. 6f, g).

The secretion of RUSH-LAMP1 from the Golgi was not altered, and neither was Golgi morphology (Fig. 6h; Supplementary Fig. 8a). However, we observed premature targeting of biosynthetic LAMPs and synaptic proteins into somatic lysosomes after 1 h (Fig. 6i–k; Supplementary Fig. 8a, b).

Together, these results show that the knockdown of VAMP4 promotes the premature targeting of LAMP compartments and synaptic proteins to somatic lysosomes at 1 h, impairing the transport of both proteins into the axon, and possibly causing synaptic protein degradation.

## RNA granules contact with and hitchhike on biosynthetic LAMP compartments along the axon

Transport of mRNA into the axon plays an important role in local axonal protein synthesis, which supports various neuron functions[45,46]. Interestingly, a recent study identified RNA-binding proteins associated to lysosomes, using LAMP1-APEX2 proteomics in human iPSC-derived neurons[27]. In addition, this and another study revealed the interaction and/or hitchhiking of RNA granules on LAMP1-positive compartments[27,47].

We also identified RNA-binding proteins, which are known constituents of RNA granules, at 4 h in our proteomics dataset, consistent with the targeting of biosynthetic LAMP to immature and mature lysosomes at 4 h (Fig. 7a; Supplementary Fig. 9a). Unexpectedly, we also observed ribonucleoprotein (RNP) complex and RNA metabolic processes in our GO analysis at 1 h (Fig. 2g), raising the possibility that RNA granules can also form contacts with and hitchhike on biosynthetic LAMP compartments.

RNA granule proteins such as FXR1, FXR2, PURA, PUM1 and PUM2 were in close proximity with biosynthetic LAMP compartments at 1 h and 4 h (Fig. 7a). We analyzed the association of the RNA granule marker FXR1 to biosynthetic LAMP compartments along the axon. After 1 h of RUSH-LAMP1 release, we observed that of all RNA granules detected along the axon, around 71% were in close proximity to biosynthetic LAMP compartments (Fig. 7b, c). On the other hand, only around 26% of all biosynthetic LAMP compartments interacted with RNA granules (Fig. 7b, c). Because biosynthetic LAMP compartments were RAB6A-positive (Fig. 5d–j), we also analyzed RAB6A−RNA granule interactions. Approximately 22% of RNA granules were in close proximity to RAB6A-positive vesicles, and 8% of RAB6A-positive vesicles were in contact with RNA granules (Fig. 7d, e). This is consistent with our results in which around 11% of LAMP1 compartments are RAB6-positive and biosynthetic in nature at the steady state (Fig. 5h–j).

To assess the dynamic interaction between RNA granules and biosynthetic LAMP compartments, we used either the PP7-PCP system[48] to visualize β-actin mRNA or GFP-FXR1. Live cell imaging of axons, after 1 h of RUSH-LAMP2A release, revealed different types of interactions: i) motile biosynthetic LAMP compartments that encounter mRNA and pause prior to resuming transport (Fig. 7f); ii) tight association of mRNA to the tips of LAMP compartments (Fig. 7g);

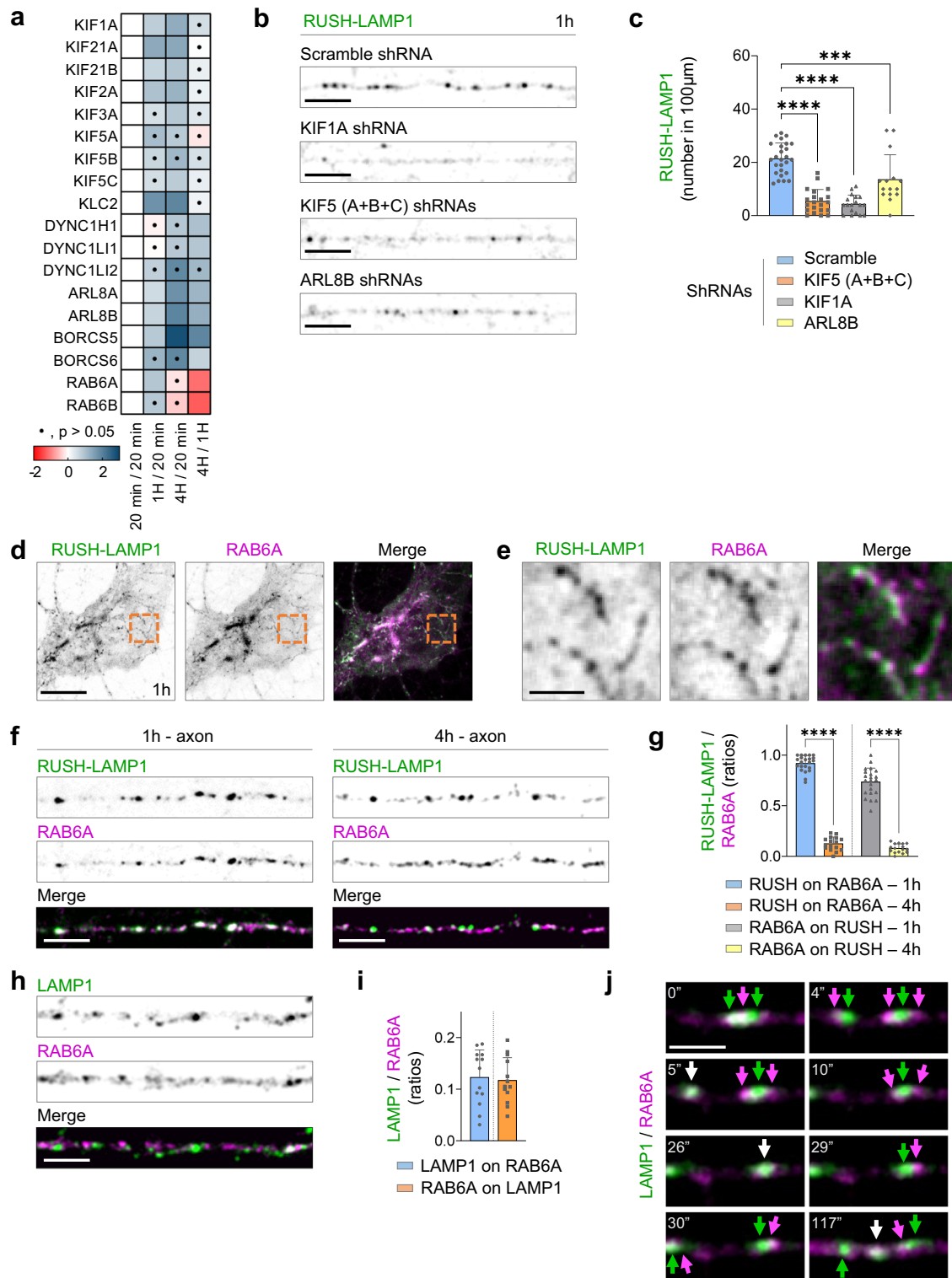

**Fig. 5 | Axonal transport of RAB6-positive biosynthetic LAMP carrier is mediated by KIF5, KIF1A and ARL8B. a** Heatmap in which the log₂ Fold Change across time points of members of the kinesin family and motor adaptors is plotted (two-sided t-test, no multiple comparison test was used). **b** Confocal images of axons of neurons expressing RUSH-LAMP1-V5 and shRNA scramble, or shRNAs against KIF5A-C, KIF1A or ARL8B. **c** Quantification of the number of LAMP1 compartments in (**b**). $n = 26, 21, 18$ and 15 neurons, respectively; each from 3 independent experiments ($N = 3$; ****$p < 0.0001$, ****$p < 0.0001$, ***$p = 0.0003$) (Scale bar, 5 μm). **d**–**f** Confocal image of the soma (**d** scale bar, 10 μm) or axon (**f** scale bar, 5 μm) of neurons expressing RUSH-LAMP1-V5 and GFP-RAB6A. **e** Magnified images of the area in (**d**) indicated by orange box (Scale bar, 2 μm). **g** Quantification of ratios for total axonal RUSH-LAMP1-V5 and RAB6A colocalizing at 1 or 4 h after release. $n = 22$ and 16 neurons; each from 3 independent experiments ($N = 3$). **h** Confocal images of axons from neurons expressing LAMP1-RFP and GFP-RAB6A. **i** Quantification of the ratios between total LAMP1 and RAB6A and colocalizing compartments, at the steady state ($n = 13$; $N = 4$ independent experiments) **j** Representative still images of the axon of a neuron expressing LAMP1-RFP and GFP-RAB6A (Scale bar, 2 μm, repeated with 4 independent experiments). Data are presented as mean values ± SD, plus individual points. Ordinary one-way ANOVA test followed by Tukey's multiple comparison test in (**c**), and Kruskal-Wallis test followed by Dunn's multiple comparison test in (**g**). See also Supplementary Fig. 6. Source data are provided as a Source Data file.

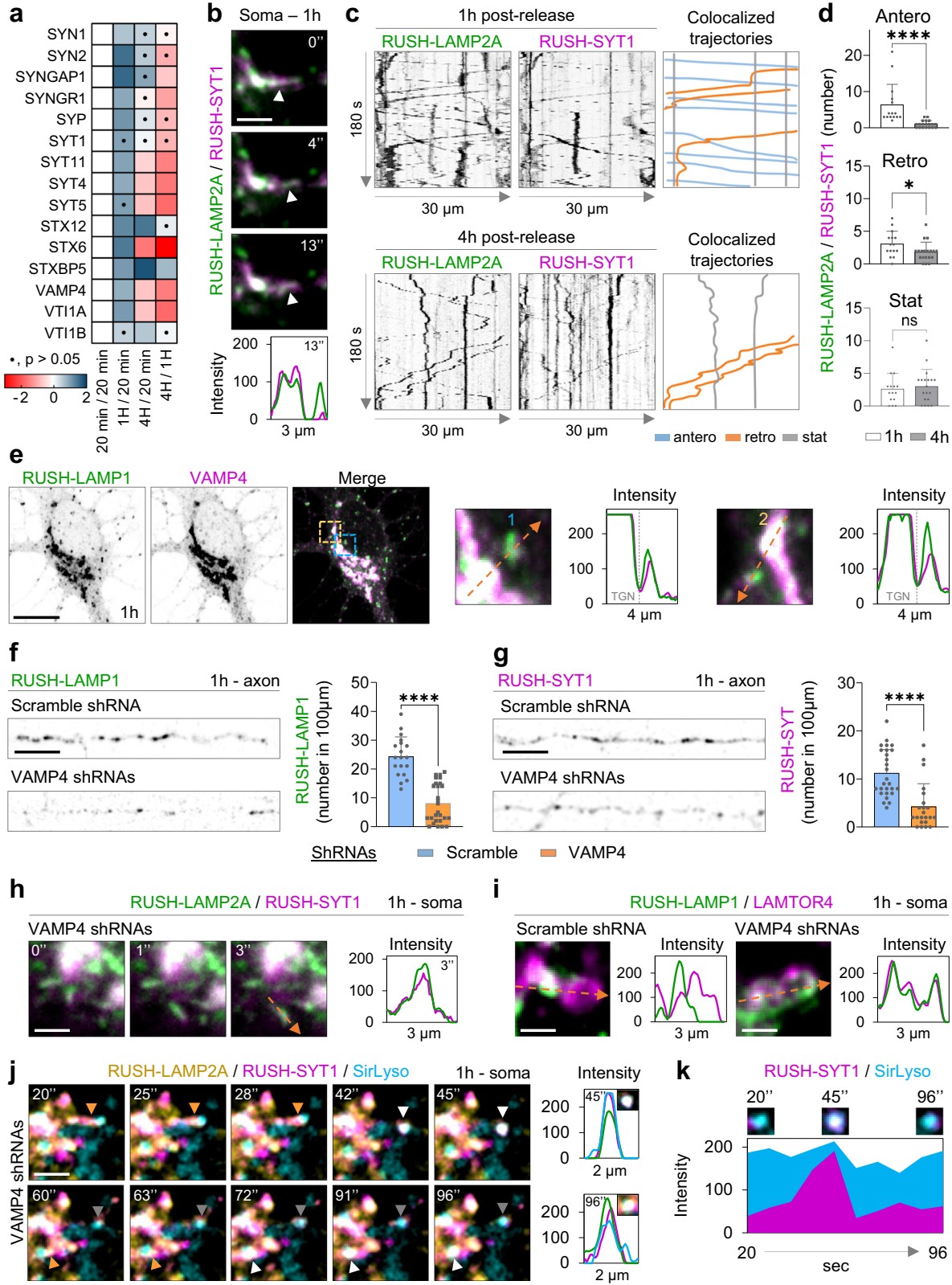

iii) fission of a RNA granule after stable association with biosynthetic LAMP compartments (Fig. 7h); and iv) hitchhiking of mRNA and RNA granules on motile biosynthetic LAMP compartments (Fig. 7i–k).

Our results suggest that the interaction of RNA granules with biosynthetic LAMP along the axon can provide an additional way for transport of mRNA for local translation.

## Discussion

In this study, we developed POTATOMap to track biosynthetic LAMP and its interactome over time, in neurons. With this approach and advanced microscopy, we identified potential neuronal domain-specific features of biosynthetic LAMP compartments, beyond axonal lysosome maintenance (Fig. 8).

**Fig. 6 | Biosynthetic LAMP and SYT1 are co-trafficked along the axon, and disruption of VAMP4 impairs their translocation to the axon. a** Heatmap in which the Log$_2$ Fold Change across time points of axonal synaptic proteins and members of the SNARE complex is plotted (two-sided t-test, no multiple comparison test was used). **b** Live neuron expressing RUSH-LAMP2A-mNG, RUSH-SYT1-Halo and scramble, imaged during 1 h of biotin addition. Still images show part of the Golgi and a budding event. Intensity profile graph in the bottom. **c** Neurons expressing RUSH-LAMP2A-mNG and RUSH-SYT1-Halo at 1 and 4 h post-release. Kymographs from live cell imaging along the axon every 1 s for 180 s are shown. Colocalized anterograde (blue), retrograde (orange) or stationary (grey) trajectories were traced on the right. **d** Quantification of the number of trajectories for 1 and 4 h. $n = 15$ and 19 neurons; each from 3 independent experiments ($N = 3$; ****$p < 0.0001$, *$p = 0.0433$, ns $p = 0.6092$) **e** Confocal images of neurons expressing RUSH-LAMP1-V5 and EGFP-VAMP4, 1 h after release. Blue and orange boxes indicate magnified areas shown on the right, with corresponding intensity profile graph. **f, g** Confocal images of neurons expressing RUSH-LAMP1-V5 (**f**) or RUSH-SYT1-mNG (**g**) plus shRNA against VAMP4, or scramble. Quantifications of the number of

RUSH-LAMP1 ($n = 19$ and 27 neurons; each from 4 independent experiments $N = 4$, ****$p < 0.0001$) and SYT1 ($n = 27$ and 22 cells; each from 3 independent experiments $N = 3$, ****$p < 0.0001$) positive compartments are shown on the right. **h** Still images from the soma of neuron expressing RUSH-LAMP2a-mNG, RUSH-SYT1-Halo and shRNA against VAMP4; control scramble in (**b**). Images show a Golgi budding event after 1 h release. Corresponding intensity profile graph on the right. **i** Neurons expressing RUSH-LAMP2A-mNG and shRNA against VAMP4 or scramble after 1 h release and immunostained for LAMTOR4 with respective intensity profile graphs are shown. **j** Neurons in (**h**) were labeled for SirLyso and imaged live after 1 h release. Still images from time points indicated in images and respective intensity profile graphs are shown. **k** Temporal intensity profile graph for RUSH-SYT1 and SirLyso from image in (**j**). Scale bars, 2 μm in (**b**), (**h–j**), 5 μm in (**f**), (**g**), and 10 μm in (**e**). Data are presented as mean values ± SD, plus individual points. Mann-Whitney test was used in (**d**), (**f**), and (**g**). See also Supplementary Figs. 7 and 8. Representative images in **b**, **e**, and **i** were repeated in at least 3 independent experiments. Source data are provided as a Source Data file.

Our POTATOMap approach provides a unique advantage to study cargo sorting and interactome from the biosynthetic pathway to destination, by combining the RUSH system with APEX2. The RUSH system allows for spatial control of synchronously secreted cargo from the ER to its desired destination. In combination with APEX2-based proximity labeling, we were able to create temporal snapshots of the interactome of LAMP at deliberately chosen times and locations from the biosynthetic pathway to the endolysosomal system. While existing LAMP-APEX proteomics have provided novel insights about lysosomes[24–32], our approach has allowed us to reveal potential key transient interactors, for an unbiased identification of key players not only regulating protein trafficking to lysosomes, but also roles of newly secreted biosynthetic compartments. Like the newly developed TransitID approach, POTATOMap enables time-resolved proteomics to study protein origin and destination[49]. However, whilst TransitID is organelle-based proteomics, POTATO-Map is protein cargo-based proteomics, offering the opportunity to identify intermediate compartments and characterize trafficking machineries.

Here, we leveraged the power of POTATOMap to reveal the nature of LAMP compartments along the axon, which has remained ambiguous for many years[10,11,50,51]. As previously reported, we observed that immature and mature lysosomes were present along the axon[10,11,50]. We found that biosynthetic LAMP and pre-existing lysosomes (immature and mature) enter the axon as separate compartments. Biosynthetic LAMP compartments were decorated with RAB6, which was particularly enriched at 1 h compared to 4 h in our proteomic data. At the steady state, we found that approximately 11% of all axonal LAMP1-positive organelles were RAB6A-positive post-Golgi carriers, in line with previous studies suggesting a heterogenous population of LAMP1-positive organelles[50,51]. This biosynthetic population could play an essential role in axonal lysosome replenishment. Indeed, we found that 2 h and 4 h after biosynthetic LAMP release, the pre-existing stationary and retrograde transported lysosomes had incorporated the biosynthetic LAMP. Together with a previous study on melanosomes, our work further consolidates the notion that RAB6-decorated LAMP carriers routing to lysosomes could be essential for proper lysosomal function[52].

We found that biosynthetic LAMP compartments required kinesin 1 and the ARL8B adaptor for their transport into the axon, like lysosomes[10]. However, kinesin 3 was also required and caused a more severe phenotype. Two possible scenarios could explain our findings. First, different biosynthetic LAMP compartments could use different motors, or second, there is a sequential step for motor-driven transport (Fig. 8). Indeed, evidence for the second scenario was previously reported, in which kinesin-1 ensures cargo entrance into the AIS and kinesin-3 provides faster transport along the axon[53]. Our data show

higher velocity of anterograde transport of RUSH-LAMP at 1 h versus 4 h, suggesting the involvement of kinesin-3 for fast transport, after the AIS, for biosynthetic LAMP compartments.

In our POTATOMap datasets, we found axonal synaptic proteins were enriched at 1 h. A previous study suggested that synaptic proteins and LAMP compartments are co-transported into the axon at the steady state, which was contradicted by another study[54,55]. Here, we show that 4 h after release, biosynthetic LAMP is mainly distributed within lysosomes, which are distinct from synaptic vesicles. However, we observed robust co-trafficking of biosynthetic proteins at 1 h into the axon. The mechanism of segregation remains unknown, but this might be tightly regulated to prevent the targeting of biosynthetic synaptic proteins into axonal lysosomes (Fig. 8).

We also found that a handful of SNAREs displayed dynamic enrichment changes over time. STX6, STX12, VTI1A, VAMP4 were enriched at Golgi and post-Golgi (1 h), and STX7, STX8 and VAMP7 were enriched at lysosomes (4 h). Interestingly we found that the known endosomal SNARE VTI1B showed no significant change over time, which could be explained by that it is a cargo co-exiting and trafficked with biosynthetic LAMP from TGN to lysosomes (Supplementary Fig. 2g). Previous independent studies have shown the importance of SNAREs in regulating synaptic vesicle availability and lysosomal function in axons[44,56]. From our data, these SNAREs were mostly enriched at 1 h during TGN exit and post-Golgi trafficking, and depletion of VAMP4 disrupted axonal targeting of biosynthetic LAMP and SYT1 compartments, leading to premature targeting of biosynthetic SYT1 to somatic lysosomes (Fig. 8). The presence of synaptic proteins in biosynthetic LAMP compartments, and the role of SNARES involved in both lysosome function and synaptic function, could suggest shared earlier mechanisms, which are essential in regulating biosynthetic protein trafficking and targeting into axon.

Unexpectedly, the second most abundant process associated with biosynthetic LAMP compartments was related to RNA metabolism. Only recent evidence has revealed the interaction of lysosomes with RNAs[27,47,57] Here we found that RNA granules contact and hitchhike on biosynthetic LAMP compartments along the axon, possibly without the risk of RNA being targeted for degradation. Thus, biosynthetic LAMP compartments could have broader roles beyond lysosome maintenance, regulating the axonal proteome by delivery of essential axonal proteins and RNA (Fig. 8).

A possible limitation of POTATOMap is the need to control proteins spilling over to other delivery routes. To counter this, we used lentivirus transduction to introduce stable expression for proteomics. Proximity labeling-based proteomic data also requires stringent validation with either imaging, as done here, or biochemistry, as identified proteins could reside on the organelle or be in close contact with neighboring compartments.

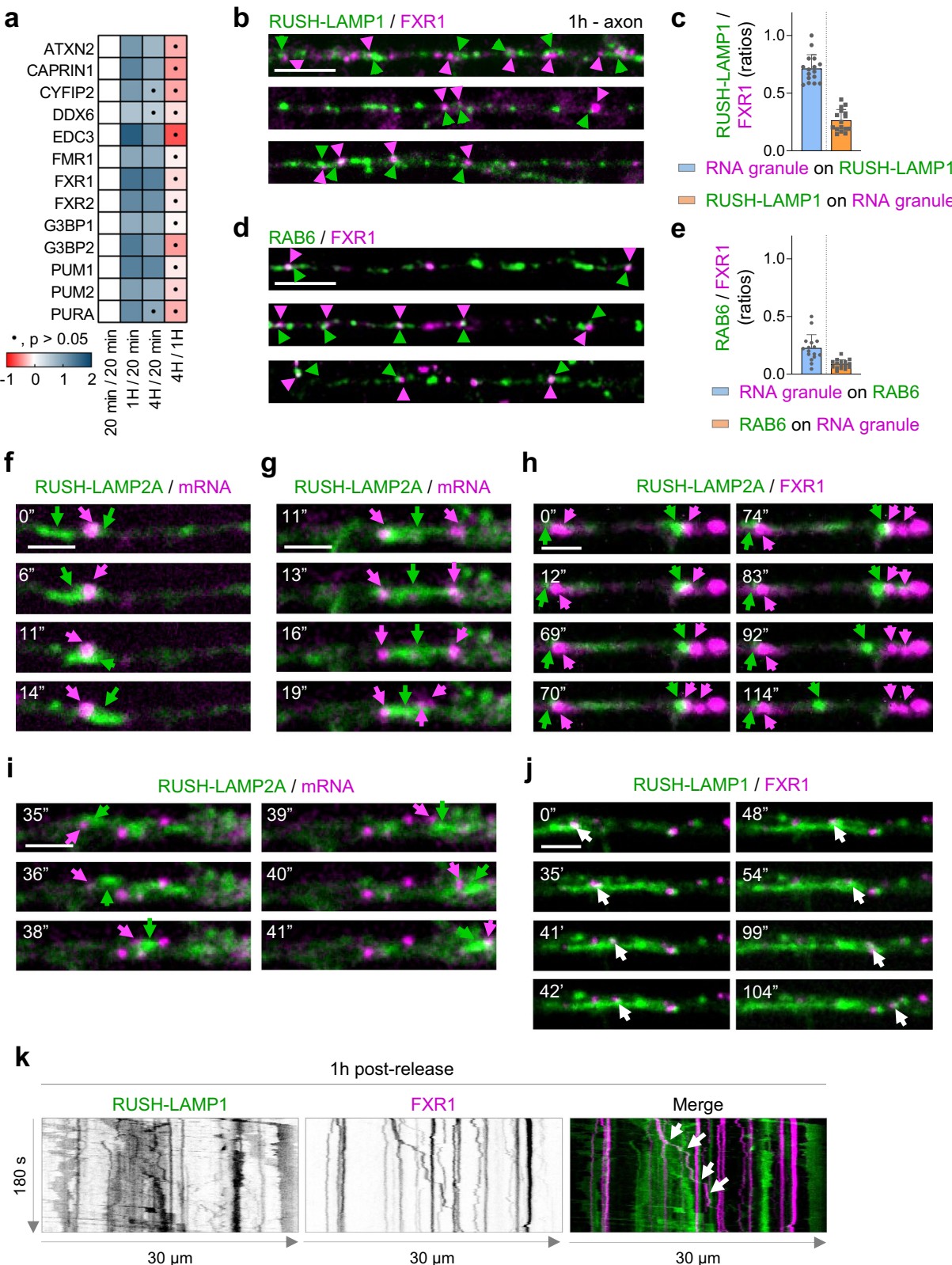

**Fig. 7 | RNA granules interact with and hitchhike on biosynthetic LAMP compartments. a** Heatmap of the Log₂ Fold Change/relative enrichment across time points of RNA granule-associated proteins (two-sided t-test, no multiple comparison test was used). **b** Confocal images of axons from neurons expressing RUSH-LAMP1-V5 and GFP-FXR1, 1 h after release. **c** Quantification of ratios between total GFP-FXR1 and RUSH-LAMP1-V5 and proximal compartments. *n* = 16; from 3 independent experiments (*N* = 3). **d** Confocal images of axons from neurons expressing GFP-RAB6 and Halo-FXR1. **e** Quantification of ratios between FXR1 and RAB6 colocalizations. *n* = 16; from 3 independent experiments (*N* = 3). **f–j** Neurons expressing RUSH-LAMP2A-mNG and actin-PP7/PCP-Halo (f, g & i) or GFP-FXR1 (h, & j), imaged every 1 s after 1 h of release. **k** Kymograph of axon shown in **j**. Scale bar, 5 μm in (**b**), (**d**), and 2 μm in (**f**), (**g**), (**h**), (**i**) & (**j**). Data are presented as mean values ± SD, plus individual points. Representative images shown in **f–j** were repeated in three independent experiments. See also Supplementary Fig. 9. Source data are provided as a Source Data file.

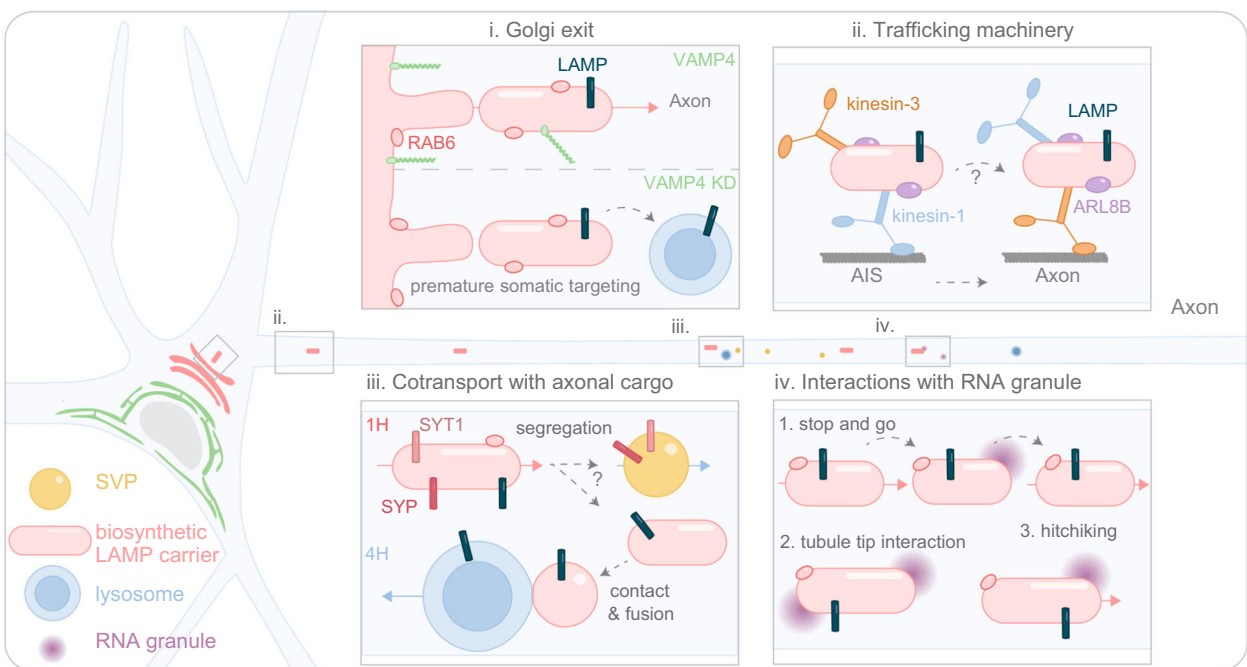

**Fig. 8 | POTATOMap reveals mechanisms of biosynthetic LAMP Golgi exit, trafficking machinery, cotransport with axonal cargo, and interactions with RNA granules.** Model of biosynthetic LAMP Golgi exit (i), trafficking machinery (ii), cotransport with axonal synaptic proteins (iii), and interactions with RNA granules (iv).

New advances in organelle-specific proteomics will allow us to perform further analysis comparing protein abundances related to a particular compartment within the biosynthetic and endosomal system, at each time point. POTATOMap can also elucidate the role of key disease-related players (Supplementary Fig. 7b). This approach will also enable exploring many other biosynthetic proteins, key players in sorting, trafficking, and targeting in health and disease.

## Methods
### Animals
All experiments were approved by the DEC Dutch Animal Experiments Committee (Dier Experimenten Commissie), performed in line with institutional guidelines of University Utrecht, and conducted in agreement with Dutch law (Wet op de Dierproeven, 1996) and European regulations (Directive 2010/63/EU). The animal protocol has been evaluated and approved by the national CCD authority (license AVD10800202216383). Female pregnant Wistar rats were obtained from Janvier, and embryos (both genders) at embryonic (E)18 stage of development were used for primary cultures of hippocampal and cortical neurons. The animals, pregnant females and embryos have not been involved in previous procedures.

### Primary neuron culture and transfection
Hippocampi or cortices from embryonic day 18 rat brains were dissected and dissociated with trypsin for 15 min and plated on coverslips coated with poly-L-Lysine (37.5 μg/ml) and laminin (1.25 μg/ml) at a density of 100,000/well (12-well plates) for live-cell imaging; or 1,000,000/well (6-well plates) for proteomics. The day of neuron plating corresponds to day-in-vitro 0 (DIV0). Neurobasal medium (NB) supplemented with 2% B27 (GIBCO), 0.5 mM glutamine (GIBCO), 15.6 μM glutamate (Sigma), and 1% penicillin/streptomycin (GIBCO) was used to maintain neurons incubated under controlled temperature and $CO_2$ (37 °C, 5% $CO_2$). Hippocampal neurons were transfected at 5 DIV using Lipofectamine 2000 (Invitrogen). Briefly, DNA (50–1000 ng/well) was mixed with 1.4 μL of lipofectamine in 100 μL of NB, for knockdown experiments 3.3 μL of lipofectamine was used. DNA/lipofectamine mixture was incubated at room temperature for 20 min,

added to neurons in NB and incubated for 45 min at 37 °C, 5% $CO_2$. Neurons were washed three times with NB and fresh biotin free medium was added. For all RUSH experiments, B27 was subjected to biotin removal using Zebra™ Dye and Biotin Removal Spin Columns (Invitrogen) according to manufacturer's protocol. Neurobasal medium containing 2% biotin free B27, 0.5 mM glutamine and 1% penicillin/streptomycin (RUSH NB medium) was used immediately after transfection. Experiments were performed either within 1-day post-transfection (DIV6-8), or for knockdown experiments 3 days post-transfection (DIV8).

### Cell culture, lentivirus packaging and lentiviral transduction
293 T cells were maintained in Dulbecco's Modified Eagles Medium (DMEM) high glucose with stable glutamine and sodium pyruvate (Capricorn Scientific) supplemented with 10% fetal bovine serum (GIBCO) and 1% Penicillin-Streptomycin (GIBCO). For POTATOMap proteomics, lentiviruses were produced by transient transfection of HEK293T cells with transfer vector containing RUSH-LAMP1-V5-APEX2, packaging vector psPAX2 and envelope vector pMD2.G with a ratio of 4:2:1 using PEI Max (Polysciences) according to standard protocols with a 3:1 PEI Max:DNA ratio for 6 to 9 h. In brief, PEI Max/DNA was mixed in fresh serum-free Opti-MEM medium (GIBCO), incubated for 20 min and added to the cells. Medium was completely removed after 6–9 h and changed to NB supplemented with 0.5 mM glutamine. Supernatants from packaging cells were collected 48–72 h post-transfection, filtered through a 0.45 μm filter, concentrated with Ultra Centrifugal Filter 100 kDa MWCO (AMICON) at 4 °C 1,000 g for 45 min. Concentrated lentivirus was resuspended in appropriate volumes of fresh RUSH NB medium and immediately used to transduce DIV4 cortical neurons. Briefly, cortical neurons were grown on 6-well plates in normal NB medium. At DIV4, neurons were briefly washed with warm NB once, and immediately transduced with RUSH-LAMP1-V5-APEX2 lentivirus in RUSH medium. Additional RUSH medium was added to a final volume of 2 mL. Cells were maintained at 37 °C and 5% $CO_2$ until further processing at DIV8. For shRNA validation, lentiviruses were produced by transient transfection of HEK293T cells with transfer vector pLKO.1 puro containing shRNA, packaging vector psPAX2 and

envelope vector pMD2.G with a ratio of 4:2:1 using PEI Max (Polysciences) according to standard protocols with a 3:1 PEI Max:DNA ratio for 6 to 9 h. In brief, PEI Max/DNA was mixed in fresh serum-free Opti-MEM medium (GIBCO), incubated for 20 min and added to the cells. Medium was completely removed after 6 to 9 h and changed to DMEM. Supernatants from packaging cells were collected 48 to 72 h post-transfection, filtered through a 0.45 μm filter, concentrated with Ultra Centrifugal Filter 100 kDa MWCO (AMICON) at 4 °C 1000 g for 45 min. Appropriate volumes of concentrated lentivirus was added to INS-1 cells.

Rat INS-1 823/3 cells (Sigma-Aldrich, scc208) were cultured in RPMI-1640 medium (Gibco 52400-025) supplemented with 10% FBS (Corning, 35079CV), 1% penicillin/streptomycin (Sigma-Aldrich, P0781) and 50uM 2-mercaptoethanol (Sigma-Aldrich M6250). Cells were maintained at 37 °C and 5% $CO_2$. Cells were transduced 1 day after plating with shRNA lentivirus and maintained for 3 days post transduction to perform knock-down of KIF1A or BORCS5.

### DNA and shRNA constructs

The following vectors were used: FUGW was a gift from David Baltimore (Addgene plasmid # 14883)[58], pLKO.1 puro was a gift from Bob Weinberg (Addgene plasmid # 8453)[59], psPAX2 and pMD2.G were gifts from Didier Trono (Addgene plasmids # 12260 and # 12259) pmScarlet-i_C1 was a gift from Dorus Gadella (Addgene plasmid # 85044)[60], H2B-mNeonGreen-IRESpuro2 was a gift from Daniel Gerlich (Addgene plasmid # 183745)[61], LAMP1-RFP was a gift from Walther Mothes (Addgene plasmid # 1817)[62], pEGFP-VAMP4 was a gift from Thierry Galli (Addgene plasmid # 42313)[63], Str-KDEL_SBP-EGFP-E-cadherin was a gift from Franck Perez (Addgene plasmid # 65286)[13], mito-V5-APEX2 was a gift from Alice Ting (Addgene plasmid # 72480)[64], pAAV hSyn GFP-FXR1 was a gift from Martin Beaulieu (Addgene plasmid # 112732)[65], LAMP1-GFP was a gift from Dr. Juan Bonifacino, GFP-RAB6A, GFP-RAB7A and GFP-RAB11A were gifts from Casper Hoogenraad[66], pAAV ORANGE Gria1-HaloTag was a gift from Harold MacGillavry, PB-Ef1a-PCP-Halo (Addgene plasmid # 198337) and PB-Ef1a-β-actin-UTR-PP7 mRNA were gifts from Michael Ward.

All the primers used in this study are provided in Source Data 3.

### For plasmids generated in this study

For generation of RUSH constructs, FUGW (Addgene plasmid # 14883) was digested with XbaI and EcoRI. Strep-KDEL-intron-IRES was amplified from Str-KDEL_SBP-EGFP-Ecadherin (Addgene plasmid # 65286) with XbaI and EcoRI sites introduced and inserted into restricted FUGW by GIBSON assembly, creating FUGW-Strep-KDEL-intron-IRES (FUGW-RUSH for simplicity). FUGW-RUSH was digested with EcoRI and BamHI to allow for downstream cloning of RUSH plasmids.

For RUSH-LAMP1-mScarleti, mScarleti was amplified from pmScarlet-i_C1 (Addgene plasmid # 85044) with an EcoRI site introduced at the 3′ end. Rat LAMP1 was amplified from LAMP1-RFP (Addgene plasmid # 1817) with the known D50E mutation corrected by PCR. A 5 amino acid linker (GSGSG) was placed after the YQTI motif of LAMP1 by PCR. Rat LAMP1 ER signal peptide sequence was amplified separately, the ER signal peptide sequence was followed with SBP amplified from Str-KDEL_SBP-EGFP-Ecadherin (Addgene plasmid # 65286) followed by a 20 amino acid linker (GSAGSAAGSGAGSAAGS-GEF) flanked by a BamHI and EcoRI site. This linker was used for the generation of other RUSH constructs to prevent misfolding (termed 20 a.a. linker for simplicity). All fragments were assembled by GIBSON assembly into restricted FUGW-RUSH between EcoRI and BamHI.

For generation of RUSH-LAMP2A-mNeonGreen, rat LAMP2A was amplified from rat cDNA library prepared from rat cortical neurons. Similar construction procedure to RUSH-LAMP1-mScarleti was performed. Briefly, rat LAMP2A ER signal peptide sequence as amplified separately and inserted with SBP and 20 a.a. linker amplified from RUSH-LAMP1-mScarleti; the rest of LAMP2A was amplified followed by

a 12 amino acid linker (GASGSAGSGASG); mNeonGreen was amplified from H2B-mNeonGreen-IRESpuro2 (Addgene plasmid # 183745) flanked by an EcoRI site. All fragments were assembled between EcoRI and BamHI of restricted FUGW-RUSH by GIBSON assembly.

To generate RUSH-CTSB-Halo, rat cathepsin B was amplified from rat cDNA library prepared from rat cortical neurons. Similar construction method as RUSH-LAMP1-mScarleti was used. In brief, rat CTSB ER signal peptide sequence was amplified separately, followed by the addition of SBP and 20 a.a. linker; the rest of CTSB was amplified by PCR followed by the addition of a 12 a.a. linker; Halo was amplified from pAAV ORANGE Gria1-HaloTag. All fragments were assembled between EcoRI and BamHI of restricted FUGW-RUSH by GIBSON assembly.

For generation of RUSH-SYT1-mNeonGreen, rat SYT1 was amplified from rat cDNA library prepared from rat cortical neurons. SBP was amplified from RUSH-LAMP1-mScarleti introduced with a Kozak sequence and GS linker in the N-terminus, SBP is flanked by a GS linker, an HA tag and a x4 GS linker added by overhang PCR. SYT1 was amplified with a 12 a.a. linker containing an EcoRI site and BamHI site (GSAGSAAGSGEF) added to the C terminus by PCR. mNeonGreen was amplified and inserted after the linker flanked by an AgeI and SwaI site. All fragments were assembled between EcoRI and BamHI restriction sites in the FUGW-RUSH vector by GIBSON assembly, where the EcoRI site was destroyed. To generate RUSH-SYT1-Halo, mNeonGreen was cut out by digesting AgeI and SwaI. Halo was amplified, flanked by the same restriction sites and inserted into between the restricted AgeI and SwaI sites of RUSH-SYT1 by GIBSON assembly.

For generation of the POTATOMap system, V5-APEX2 was amplified from mito-V5-APEX2 (Addgene plasmid # 72480). LAMP1 was amplified from RUSH-LAMP1-mScarleti after the 20 a.a. linker, the original GSGSG linker was replaced with a new 12 amino acid linker (GSAGSAAGSGEF). RUSH-LAMP1-mScarleti was restricted with the two EcoRI sites flanking the LAMP1-mScarleti gene, leaving SS-SBP-20 a.a. linker in the FUGW-RUSH vector. Fragments were assembled between the digested EcoRI sites by GIBSON assembly (construct termed RUSH-LAMP1-V5-APEX2). To generate RUSH-LAMP1-V5, same construction procedure was performed but omitting APEX2 during amplification from mito-V5-APEX2 (Addgene plasmid # 72480).

For generation of Halo-(2xFKBP)-FXR1, FXR1 was amplified from pAAV-hSyn-GFP-msFxr1 (Addgene plasmid #112732) and Halo was amplified from pHalo-ZBP1 (Koppers lab). Fragments were subcloned into lentiviral vector containing a UbC promoter. The plasmid contains a 2xFKBP sequence for other purposes not used in this study.

The following sequences for rat-shRNAs inserted to pLKO.1-puro were used in this study: scramble shRNA (5′-GATGAAATATTCCGCAA GTAA-3′) was used for experiments related to KIF1A, KIF5A/B/C, ARL8B and BORCS5 knockdown. KIF5A-shRNA (5′-GAGACATCTTCAACCAC AT-3′), KIF5B-shRNA (5′-TGGAGGGTAAACTTCATGA-3′), KIF5C-shRNA (5′-TGAGATCTACTTGGACAAA-3′) and KIF1A-shRNA (5′-CACGCCGTC TTCAACATCA-3′) were validated in Farias et al., 2019[40]. ARL8B-shRNA#1 (5′-ACCGAGAGATCTGCTGCTA-3′) and ARL8B-shRNA#2 (5′-GAACCTGTCTGCTATTCAA-3′) were validated in Hummel et al., 2021[67]. BORCS5-shRNA#1 (5′-GCGAATCAAAGAGATGGATCT-3′) and BORCS5-shRNA#2 (5′-ACTTTGCATCAAGAGTTATTT-3′) against rat BORCS5 were validated by qPCR. Scramble shRNA (5′- GCTTCAAT AACTAAAGATA-3′) was used as control for VAMP4 knock down. VAMP4-shRNA (5′-GGACCATCTGGACCAAGATTT-3′) was validated in Bakr et al., 2021[68]. All shRNAs have been validated in this study by using immunofluorescence or quantitative PCR.

The following sequences were used in this study to validate KIF1A and BORCS5 shRNA knockdown efficiency: KIF1A primer pair 1 (5′-CAC TGACACCAACACTGTGC-3′ and 5′- GTGGAGACTGGACACAGAGG-3′), KIF1A primer pair 2 (5′-AAGGATGAGGTGACCAGGCT-3′ and 5′-TCAATT TGGGTCCTCCAGGC-3′), BORCS5 primer pair 1 (5′-CTGAACGCCTCAG TGACTCC-3′ and 5′-TGAGAGCCCTGAGCTACCAC-3′), BORCS5 primer

pair 2 (5′-CATGGGCAGCGAGCAGAG-3′ and 5′-TCCATCTTGGCCCGAT GTTT-3′). Primers targeting GAPDH were used as internal control, GAPDH primer pair (5′-CAACTCCCTCAAGATTGTCAGCAA-3′ and 5′-GGCATGGACTGTGGTCATGA-3′).

## Antibodies and reagents

The following primary antibodies were used in this study: rabbit anti-LAMTOR4 (Cell Signaling, clone D6A4V, Cat# 12284S, RRID: AB_2797870, 1/500), mouse anti-STX6 (BD Biosciences Cat# 610635, RRID:AB_397965, 1/100), rabbit anti-VAMP4 (Synaptic Systems, Cat# 136002, RRID:AB_887816, 1/100), mouse anti-V5 (Thermo Fisher Scientific Cat# R960-25, RRID:AB_2556564, 1/1000 for IF and WB), mouse anti-Pan-Neurofascin external (clone A12/18; UC Davis/NIH NeuroMab, Cat# 75-172, RRID: AB_2282826, 0.18 mg/ml), in-house rabbit anti-TRIM46 (1/1000), mouse anti-VTI1B (BD Biosciences Cat# 611404, RRID:AB_398926, 1/250), rabbit anti-GM130 (Abcam Cat# ab52649, RRID:AB_880266, 1/800), rabbit anti-ARL8B (Proteintech Cat#13049-1-AP, RRID:AB_2059000, 1/500), rabbit anti-KIF5A (Abcam Cat# ab5628, RRID:AB_2132218, 1/100), rabbit anti-KIF5B (Abcam Cat# ab5629, RRID:AB_2132379, 1/100), rabbit anti-KIF5C (Abcam Cat# ab5630, 1/100).

The following secondary antibodies were used in this study: rabbit anti-mouse immunoglobulins/HRP (Agilent Cat# P0260, RRID:AB_2636929, 1/10000), goat anti-rabbit IgG (H + L) Highly cross-absorbed secondary antibody Alexa Fluor 405 (Thermo Fisher Scientific Cat# A-31556, RRID:AB_221605, 1/1000), donkey anti-rabbit IgG (H + L) Highly cross-absorbed secondary antibody Alexa Fluor 647 (Thermo Fisher Scientific Cat# A-31573, RRID:AB_2536183, 1/1000), goat anti-rabbit IgG (H + L) highly cross-absorbed secondary antibody Alexa Fluor 568 (Thermo Fisher Scientific Cat# A-11036, RRID:AB_10563566, 1/1000), goat anti-mouse IgG1 cross-absorbed secondary antibody Alexa Fluor 594 (Thermo Fisher Scientific Cat# A-21125, RRID:AB_2535767, 1/1000), goat anti-mouse IgG2a cross-absorbed secondary antibody Alexa Fluor 594 (Thermo Fisher Scientific Cat# A-21135, RRID:AB_2535774, 1/1000).

Other reagents used in this study were Streptavidin Alexa Fluor-555 conjugate (Thermo Fisher Scientific Cat# S21381, 1/1000), IRDye 800CW Streptavidin (LI-COR Biosciences, Cat#926-32230, 1/10000), antibody labeling kit Mix-n-Stain CF640R (Biotium), Lipofectamine 2000 (Invitrogen, Cat#1639722), SiR-lysosome kit (Spirochrome, Cat# SC012), biotin-phenol (Iris Biotech, Cat#LS.3500); $H_2O_2$ (Sigma-Aldrich, Cat#H1009), biotin (Sigma-Aldrich, Cat#B4501), Saponin (Sigma, Cat#47036), Sodium Azide (Merck, Cat#K43547688), Sodium L-ascorbate (Sigma, Cat#0000374819), Trolox (Sigma, Cat#238813), JFX554 was kindly provided by the Lavis Lab (Janelia).

## RNA extraction, reverse transcription and quantitative PCR

Rat INS-1 was used to validate knockdown efficiency of KIF1A and BORCS5 shRNA. Total RNA from transduced rat INS-1 was extracted using RNeasy Mini kit (Qiagen, 74104). cDNA was reverse-transcribed using SuperScript III First-Strand Synthesis System (Invitrogen, 18080-051) according to manufacturer's instructions. Quantification of target genes was done by qPCR using Power SYBR Green PCR Master Mix (Applied Biosystems, 4367659) and ViiA 7 Real-Time PCR System (Applied Biosystems). Data was extracted from ViiaA 7 Real-Time PCR system using QuantStudio Software V1.3 (Applied Biosystems).

## Live-cell imaging

For live-cell imaging experiments, an inverted microscope Nikon Eclipse Ti-E (Nikon), equipped with a Plan Apo VC ×100 NA 1.40 oil objective (Nikon), a Yokogawa CSU-X1-A1 spinning disk confocal unit (Roper Scientific), a Photometrics Evolve 512 EMCCD camera (Roper Scientific) and an incubation chamber (Tokai Hit) mounted on a motorized *XYZ* stage (Applied Scientific Instrumentation) was used. MetaMorph (Molecular Devices) version 7.10.2.240 software was installed for controlling all devices. Coverslips mounted in a metal ring and supplemented in the original medium from neurons were imaged in an incubation chamber that maintains optimal temperature and $CO_2$ (37 °C and 5% $CO_2$). To visualize multiple fluorescently labelled proteins and probes, sequential imaging was used, and each laser channel was exposed for 150−200 ms. Neurons were imaged every 1 s for 180 s or 1 min for 60 min. To identify the axon, neurons were either co-transfected with BFP fill and identified by morphology or incubated with a CF640R-conjugated antibody against the AIS protein neurofascin (NF-640R) for 30 min before live-cell imaging[69]. To label Halo-Tag, JFX554 was incubated (100 nM in RUSH medium) for 15 min prior to biotin release of RUSH cargoes, washed three times with warm NB, and returned to original RUSH medium. Total time and intervals of imaging acquisition for each experiment are depicted in each figure legend.

## Live mature lysosome labelling

To probe for cathepsin D activity for live-cell imaging, DIV6-8 hippocampal neurons were incubated with SirLyso (100 nM in RUSH medium; Spirochrome) for 15 min at 37 °C and 5% $CO_2$. Cells were washed twice with NB and supplemented with the original medium and imaged immediately.

## Immunofluorescence staining and imaging

Neurons were incubated at room temperature with 4% paraformaldehyde supplemented with 4% sucrose in PBS for 10 min for fixation. Cells were permeabilized with 0.2% Triton X-100 in PBS supplemented for 10 min followed by blocking with 0.2% porcine gelatin in PBS for 1 h at room temperature. For experiments involving RAB GTPase imaging, cells were permeabilized and blocked with 0.05% saponin supplemented with 0.2% porcine gelatin in PBS for 1 h at room temperature. Cells were incubated with primary antibodies overnight at 4 °C, washed three times with PBS, incubated with secondary antibodies for 45 min at room temperature and washed three times with PBS. Cells were mounted in Fluoromount-G Mounting Medium (ThermoFisher Scientific). Cells were imaged using a confocal laser-scanning microscope (LSM900, with Zen (blue edition) imaging software version 3.7.97.07000 (Zeiss)) equipped with Plan-Apochromat ×63 NA 1.40 oil DIC objective. For knockdown experiments, only cells displaying continuous labeling of the cytosolic fluorescent protein (fill) along the somatodendritic and axonal domains were imaged.

## Protein Origin, Trafficking And Targeting to Organelle Mapping (POTATOMap) sample preparation

DIV8 neurons transduced with lentivirus at DIV4 stably expressing RUSH-LAMP1-V5-APEX2 (termed POTATOMap) were incubated with RUSH media supplemented with 500 μM biotin-phenol (IrisBio-tech) at 37 °C for 20 min before addition of 1 mM $H_2O_2$ (Sigma) at room temperature for 1 min to trigger peroxidase activity. Biotinylation was immediately quenched by two washes in ice-cold quencher solution (10 mM sodium azide, 10 mM sodium ascorbate and 5 mM Trolox in HBSS) and incubated 10 min on ice in azide-free quencher solution. Cells were scrapped and pulled according to their respective conditions. Pellets were stored at −80 °C until all samples were harvested. Immunofluorescence was performed in the same batch of cortical neurons culture grown on 12-well plate to validate transduction efficiency as described in methods. Cell pellets were lysed with freshly made quenching-RIPA buffer (150 mM NaCl, 50 mM Tris-HCl pH7.4, 0.1% SDS, 0.5% sodium-deoxycholate, 1% Triton X-100, 10 mM sodium azide, 10 mM sodium ascorbate, 5 mM Trolox and 1× protease inhibitors (Roche)) on ice for 30 min. Lysates were cleared by centrifuging at 16,000 × g 4 °C, supernatants were incubated overnight with pre-equilibrated Pierce magnetic streptavidin beads (Invitrogen, Cat#88817) on a rotor at 4 °C. Beads were washed three times with detergent-free quenching-RIPA buffer (150 mM NaCl, 50 mM Tris-HCl pH7.4, 10 mM sodium azide, 10 mM sodium ascorbate, 5 mM Trolox)

and three times freshly prepared 3 M Urea buffer (Urea in 50 mM ammonium bicarbonate). Beads were resuspended in 50 μL of 3 M Urea buffer and reduced with 5 mM TCEP (Sigma) at room temperature for 30 min, alkylated with 10 mM IAA (Sigma) in the dark at room temperature for 20 min and quenched with 20 mM DTT (Sigma). Samples were washed three times with 2 M Urea buffer and resuspended in 50 μL 2 M Urea buffer. Suspended beads were first incubated with 1 μg LysC for 4 hours at room temperature, followed by adding 1 μg Trypsin for digestion at 37 °C overnight. Peptides were collected by combining digested supernatant with two subsequent 50 μL 2 M Urea buffer washes and immediately acidified with 1% trifluoroacetic acid. Digested peptides were desalted on Sep-Pak C18 Cartridges (Waters) and vacuum concentrated for storage until subsequent MS analysis.

## Immunoblotting

Lysates were prepared as described above prior to beads incubation. 5% of lysates were taken for immunoblotting. Protein lysates were resolved by SDS-PAGE on a 10% Bis-Acrylamide (Bio-Rad) gel and transferred to a PVDF membrane (Bio-Rad). Membranes were blocked in 5% skimmed milk in TBS-T, washed three times with TBS-T. Membranes were incubated overnight at 4 °C with primary antibodies in antibody buffer (3% BSA in TBS-T). After three washes with TBS-T, membranes were incubated with anti-mouse HRP secondary antibody in antibody buffer for 45 min at room temperature and washed three times with TBS-T. Membranes were then incubated in Clarity Western ECL Substrate (Bio-Rad, Cat#1705060) and developed using ImageQuant 800 (AMERSHAM). To visualize biotinylation, membranes were washed three times in TBS-T and incubated with IRDye 800CW Streptavidin for 45 min at room temperature. Membranes were washed two times in TBS-T and once in TBS and developed on an Odyssey CLx imaging system (LICOR) with Image Studio version 5.2.

## MS and data analysis

All samples were reconstituted in 0.1% formic acid and analyzed on a Orbitrap Exploris 480 mass spectrometer (Thermo Fisher Scientific, San Jose, CA, United States) coupled to an UltiMate 3000 UHPLC system (Thermo Fisher Scientific, San Jose, CA, United States). Peptides were loaded onto a trap column (C18 PepMap100, 5 μm, 100 Å, 5 mm × 300 μm, Thermo Fisher Scientific, San Jose, CA, United States) with solvent A (0.1% formic acid in water) at 30 μl/min flowrate and chromatographically separated over the analytical column (Poroshell 120 EC C18, Agilent Technologies, 50 μm × 75 cm, 2.7 μm) using 180 min gradient at 300 nL/min flow rate. The gradient proceeds as follows: 9% solvent B (0.1% FA in 80% acetonitrile, 20% water) for 1 min, 9–13% for 1 min, 13–44% for 155 min, 44–55% for 5 min, 55–99% for 5 min, 99% for 3 min, and finally the system equilibrated with 9% B for 10 min.

The mass spectrometers were used in a data-dependent mode, which automatically switched between MS and MS/MS. After a survey MS scan ranging from 375 to 1600 m/z with 14 s dynamic exclusion time, the most abundant peptides of 120 m/z or higher were subjected to high energy collision dissociation (HCD) for further fragmentation using a 1.4 m/z isolation window. MS spectra were acquired in high-resolution mode (R > 60,000), whereas MS2 was in high-sensitivity mode (R > 15,000) and 28% normalized collision energy.

For data analysis, raw files were processed using MaxQuant's (Version 2.0.1.0)[70]. Andromeda search engine in reversed decoy mode based on rat reference proteome (Uniprot-FASTA, UP000002494, downloaded March 2023) with an FDR of 0.01 at both peptide and protein levels. Digestion parameters were set to specific digestion with trypsin with a maximum number of 2 missed cleavage sites and a minimum peptide length of 7. Oxidation of methionine and amino-terminal acetylation were set as variable and carbamidomethylation of cysteine as fixed modifications. The tolerance window was kept at default. Label-free quantification is applied (minimum ratio count set to 2), and a total of 2 biological replicates with 2 technical replicates were analyzed. The resulting protein group file was processed using Perseus (version 1.6.15.0)[71]. Briefly, common contaminants, reverse, site-specific identifications were filtered out. Proteins with peptide count and unique peptide count <2 were excluded. Each condition was then separated. Background control groups (w/o $H_2O_2$) with less than 1 valid value would be excluded, while experimental group (with $H_2O_2$) with less than 3 valid values would be excluded. The remaining hits were subjected to imputation from normal distribution set separately for each column with width 0.3 and down shift 1.8. Each experimental group was joined back to their respective background control. Hits that were shared between experimental group (with $H_2O_2$) and their respective background control (w/o $H_2O_2$) of the same timepoint were tested using unpaired two sample t-test ($p$ value < 0.05, $\log_2$ fold change ≥1); hits that were unique to experimental group of the specific timepoint were joined back to filtered experimental group. Filtered experimental group from each timepoint was then tested against each other (1 h/20 min; 4 h/20 min; 4 h/1 h) using unpaired two sample t-test ($p$ value < 0.05, $\log_2$ fold change ≥1 or ≤−1). Volcano plots were generated using VolcaNoseR[72]. To identify the temporal signature of a particular protein, venn-diagram was constructed by analyzing only the significantly changing proteins in their respective test condition ($p$ value < 0.05, $\log_2$ fold change ≥1 or ≤−1). Hits unique to a specific timepoint were not depicted unless specified. DAVID (version 7.0) and Gene Ontology were then used for functional enrichment analysis, and STRING analysis was performed for each specific group using Cytoscape version 3.7.2. BP, biological process; CC, cellular component; MF, molecular function; KEGG; UPKW, Uniprot key words. All terms were filtered (FDR < 0.01) and dot plots were generated by plotting each selected term according to their annotation, number of proteins categorized in that term as size, and a color gradient was used to depict the $p$ value.

Figure 3a was generated by grouping proteins according to their respective biological processes or molecular functions. Dotted lines were used to depict physical interactions according to STRING analysis. Proteins were color coded according to their respective enrichment in each comparison between timepoints.

Heat maps were generated by selecting specific hits and plotted according to their respective fold change with $p$ value > 0.05 denoted. Detailed $p$ value can be found in Source Data 1.

Venn diagram comparison between POTATOMap 4 h (276 proteins, D ≥ 1, $p$ < 0.05) and selected lysosome proteomics dataset in Supplementary Fig. 3 was performed as follows: (a) Significant proteins (105) from Liao et al.[27] were selected and compared to POTATOMap 4 h; (b) endogenously knocked-in LAMP1-APEX2 data from Frankenfield et al.[31] were first filtered against their in-house no APEX2 background, filtered hits were then filtered against their in-house NES-APEX2 control yielding a total of 108 proteins to compare to POTATOMap 4 h; (c) 1198 proteins in Neuro2a Lyso-IP proteomics from Krogsaeter et al.[32] based on their significance annotated by the authors and compared to POTATOMap 4 h; (d) Mouse Lyso-IP data from Laqtom et al.[25] were first filtered for $\log_2$Fold Change >1 and known lysosomal protein was set to true, yielding 247 significant proteins. Selected candidates were compared to POTATOMap 4 h. DAVID analysis was performed as mentioned above, where all terms were filtered (FDR < 0.01). Source data is available at Source Data 1.

Identification and comparison between RNA granule markers was performed using RNA granule database[73]. Briefly a list of genes was curated from Markmiller et al. and Zhang et al., genes scored as tier one were used for comparison to generate a venn-diagram (Supplementary Fig. 9a)[74,75]. Source data is available at Source Data 1.

## Image analysis and quantification

Fluorescence line intensity plots: co-distribution of different proteins was analyzed using Image *J*. Plot profiles were generated using a line traced along specified markers. Length of traced profile line is indicated in each intensity plot.

Kymograph analysis: Kymographs were generated from live cell images using Image *J*. Segmented lines were drawn and straightened along the axon identified by morphology using a fill and/or using the AIS marker NF-CF640R. Straightened axons were re-sliced followed by z-projection to obtain kymograph. A random segment of 30μm was cropped out for analysis. Anterograde movements were oriented from left to right in all kymographs. Time of recording and length of segments are indicated in each kymograph (Fig. 4a, d–f; Fig. 6c; Fig. 7k). The number of events for anterograde, retrograde, stationary and total number of RUSH-LAMP2A, LAMP1-GFP, SirLyso and RUSH-SYT1 were obtained from three independent experiments. To measure velocity of trajectories (Supplementary Fig. 5g), straight line ROIs were manually drawn along mobile segments of individual traces. The angle of the lines was used to calculate the average velocity of the mobile segments. Formula is available in Source Data 2.

Mander's colocalization analysis: We analyzed the colocalization between RUSH-LAMP1 and the markers STX6, RAB11, RAB7, or LAMTOR4 in the soma using Image *J*. The soma was selected and regions outside were cleared. Then, the Just Another Colocalization Plugin (JACoP)[76] was used to calculate the Mander's coefficients for the overlap between two channels (Supplementary Fig. 4, left graphs).

Quantifying the number of biosynthetic compartments, STX6, RAB11, RAB7, RAB6A, lysosomes and RNA granules in axon: Axons were identified by morphology using a fill and/or with the AIS marker TRIM46. The number of biosynthetic compartments, STX6, RAB11, RAB7, RAB6A, lysosomes and RNA granules were analyzed using Image *J* (Fig. 5b, f, h; Fig. 6f, g; Fig. 7b, d; Supplementary Fig. 3a–d; Supplementary Fig. 6h). To quantify the number of axonal biosynthetic LAMP1 or biosynthetic SYT1 after shRNA depletion of targeted proteins (Fig. 5b; Fig. 6f, g; Supplementary Fig. 6h), segmented lines were drawn approximately 30μm away from the cell body along the axon and straightened. 100μm axon-length was used for quantification. Number of puncta was counted manually in each straightened axon. To quantify the distribution of biosynthetic LAMP1 compartments and steady state lysosomes with GFP-RAB6A (Fig. 5f, h), and biosynthetic LAMP compartments with STX6, RAB11, RAB7 and LAMTOR4 (Supplementary Fig. 3a–d), segmented lines were drawn approximately 30μm away from the cell body along the axon for 100μm and straightened for quantification. Total number of puncta was counted manually in individual channels. Merged channels were used to quantify colocalization and a minimum of 90% pixels from both channels (4–8 pixels depending on vesicle sizes) overlap would be considered colocalization. Briefly, complete colocalization (white) between both channels was considered as colocalization, while partial colocalization (green and magenta) was not considered as co-distribution. Ratio was calculated by dividing the number of colocalized puncta to the total number of the target compartment. To quantify the proximity of GFP-FXR1 and biosynthetic LAMP1 (Fig. 7b) and Halo-FXR1 and GFP-RAB6 (Fig. 7d), segmented lines were drawn along the axon for 100μm and straightened for quantification. Total number of puncta was counted manually in individual channels. Merged channels were used to quantify proximity. No specific strategy for randomization and/or stratification was employed. Data was analyzed by at least two people using Image *J*.

Validation of shRNA efficiency: shRNA efficiency targeting KIF5A-C, ARL8B and VAMP4 was quantified by immunofluorescence intensity using Image *J*. Intensity was measured at the axon tip endogenously stained for KIF5A-C and ARL8B by tracing with a cytosolic fill (Supplementary Fig. 6a–d). VAMP4 intensity was measured in the soma (Supplementary Fig. 7c). Then, each value was normalized to the mean intensity of the control (scrambled).

## Statistics and reproducibility

Data processing and statistical analysis were performed using Microsoft Excel, GraphPad Prism (version 9.5.1), MaxQuant (Version 2.0.1.0) and Perseus (version 1.6.15.0). Unpaired *t*-tests, Mann-Whitney tests, Kruskal-Wallis test followed by Dunn's multiple comparison test, ordinary one-way ANOVA tests followed by Tukey's multiple comparison test were performed for statistical analysis. Significance as determined as following: ns- not significant, $*p < 0.05$, $**p < 0.01$, $***p < 0.001$, and $****p < 0.0001$. The assumption of data normality was checked using D'Agostino–Pearson omnibus test. The statistical test performed, number of cells ($n$) and independent experiments ($N$) are indicated in figure legends. Exact $p$ values are reported in figure legends and Source Data. No statistical method was used to pre-determine sample size, and no data were excluded from the analyses. The experiments were not randomized and investigators were not blinded to allocation during experiments and outcome assessment. See also Source Data 1 and 2.

## Reporting summary

Further information on research design is available in the Nature Portfolio Reporting Summary linked to this article.

# Data availability

Plasmids generated in this study can be obtained from Addgene (#229524-228527). The POTATOMap mass spectrometry proteomics data have been deposited to the ProteomeXchange Consortium via the PRIDE partner repository with the dataset identifier PXD056787. The processed POTATOMap proteomics data generated in this study are provided in Source Data 1. Exact $p$ values are provided in Source Data. Source Data 1 for relevant proteomics analysis; Source Data 2 for relevant imaging analysis; Source Data 3 for relevant oligos. All data are freely available upon request. Source data are provided with this paper.

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

## Acknowledgements
We thank Ha Nguyen (Farias lab) for providing RUSH-SYT1 constructs. We thank Mirjam Damen and Cristina Trueba Sanchez (Utrecht University) for their advice on proteomic sample preparation, data acquisition, and analysis. We thank Professor Dr. Judith Klumperman (University Medical Center Utrecht, NL) for providing critical feedback and discussion during the project. This work was supported by the European Research Council (ERC-StG 950617), the Netherlands Organization of Scientific Research (0.16.VIDI.189.019) to G.G.F., and the European Union's Horizon 2020 research and innovation program under the Marie Skłodowska-Curie grant agreement (ITN-SAND 860035) to GGF, and the Netherlands Organization of Scientific Research (OCENW.KLEIN.236) to G.G.F. & J.K.

## Author contributions
C.H.L. designed and performed experiments, analyzed data, and wrote the manuscript; N.K. designed and performed experiments, analyzed data, and wrote the manuscript; N.Ö. analyzed data related to RUSH-LAMP1 and FXR1; D.T.M.N. performed experiments and analyzed qPCR data related to shRNA knockdown; M.K. performed experiments, provided feedback, and edited the manuscript; H.P. provided mass spectrometry training, proposed experiments and provided feedback; A.F.M.A. provided mass spectrometry expertise and feedback, discussed data and edited the manuscript; G.G.F. designed experiments, analyzed data, supervised the research, coordinated the study, and wrote the manuscript.

## Competing interests
The authors declare no competing interests.
