## [Peer Review file · Nature Communications]

Spatiotemporal proteomics reveals the biosynthetic lysosomal membrane protein interactome in neurons

Corresponding Author: Dr Ginny Farías

Version 0:

Reviewer comments:

Reviewer #1

(Remarks to the Author)

This manuscript described the dynamic biosynthetic LAMP trafficking and interactome in neurons by combining RUSH microscopy and APEX proteomics strategies, namely, POTATOMap. Using this method, authors were able to track the movement of LAMP through the ER, Golgi and lysosome over different time points. Interestingly, this method found that immature and mature lysosomes enter the axon as separate compartments with distinct protein features, and that RNA granules traffic along the axons by attaching to biosynthetic LAMP compartments.

Overall, this is an elegant study that introduced a new methodology and revealed important roles of axonal lysosomes in delivery newly synthesized axonal synaptic proteins via kinesin and adaptor proteins and interactions with RNA granules. But the unique features/advantages of POTATOMap are not clearly discussed, and how are the proteomics results directly compared with the existing Lyso-IP or Lyso-APEX approaches are not provided. The authors also need to be careful to avoid overdrawing conclusions from the microscopy data. My main comments are listed below:

Major Comments:

- Since the RUSH system is frequently used in this manuscript, including a brief description of how RUSH work would be helpful in the introduction section.
- Why combining RUSH and proximity labeling provide unique advantages is not clear.
- In line 155, the authors mentioned several overlapping proteins from POTATOMap with previously published lyso-IP and lyso-proximity labeling proteomics. It would be helpful to provide a direct comparison of the proteome coverage (for example venn diagrams) among POTATOMap, Lyso-IP, and Lyso-APEX proteomics.
- The authors also missed several key citations from recent years that used LAMP1-proximity labeling to study lysosome interactome. They should be cited in this manuscript: Bhattacharya et al., Nat Cell Biol, 2023 (<https://pubmed.ncbi.nlm.nih.gov/37024685/>); Hasan et al., Mol Neu. 2023 (<https://pubmed.ncbi.nlm.nih.gov/37974165/>); Tan et al, Nature, 2022(<https://pubmed.ncbi.nlm.nih.gov/36071159/>)
- In Figure 2a, the authors show that POTATOMap was conducted using a 20 min, 1 hr, and 4 hr timepoint. What's the rationale of selecting these time points? What would happen after 4 hours?
- In Figure 5, line 238, the authors estimated around 15% of LAMP1-positive organelles correspond to biosynthetic LAMP compartments. Similarly in Figure 7, line 307-309, the authors estimated the percentage of LAMP compartments. How are these percentage values calculated and quantified? Do they have biological replicates to support the reproducibility?
- Overexpressing LAMPs have been previously shown to cause mislocalization to the plasma membrane. Have the authors observed this possible problem?

Minor comments

- Please make sure all abbreviations are spelled out in the manuscript. For example, TGN has been used multiple times in

the manuscript, does it mean trans-Golgi network?

•Please carefully read through the entire manuscript to avoid typos and inconsistency of the use of words, symbols, and capitalization.

Reviewer #2

(Remarks to the Author)

In this study, the authors developed a novel system they dubbed POTATOMap. The system combines the “retention using selective hook (RUSH)” to synchronize the “releasing” of proteins from the ER and the APEX-based proximity labeling-based proteomics to reveal the interactomes at various time points after protein releasing. By applying the system to the lysosome-related proteins LAMPs in cultured neurons, the authors “released” LAMP proteins and monitored their interactomes at various time points as they are delivered to lysosomes. The studies reveal the gradual changes of the interactome environment and provide a snapshot of the system with an unprecedented temporal resolution. Several hundred proteins are discovered that are at some point within the vicinity of the LAMPs.

There are some novel interesting findings in the study. For example, the authors discovered that at ~1 hr after the “release”, the LAMP interactome includes some of the usual suspects such as those involved in Golgi- and endosome-transport, but also proteins involved in RNA processing and synaptic functions, before lysosome “functional” proteins such as mTOR were found ~4 hr after the “release”. Using live cell imaging and shRNA knock-down, the authors also demonstrated the significance of some of the components they found in the interactomes in functions such as organelle transport.

The studies described in the article are comprehensive and multi-disciplined. Many of the findings are novel and reveal new lysosome biology. The studies will be useful for researchers in the field of lysosome biology. Perhaps more importantly, some of the “-omics” findings in the studies will be the basis for future detailed mechanistic studies.

The studies are well-designed. The presentation is clear and easy to understand. I have only 3 minor comments.

1) The authors claim that they “map the endogenous interactome”. The interactomes, are, however, from that of exogenously expressed LAMP that is fused with other proteins (RUSH-LAMP1-V5-APEX2). It should be noted that the interactomes are not necessarily the same as those of the endogenous LAMP, especially if the protein is over-expressed. At minimum, it'd be useful if experiments were performed to compare the protein level of the exogenously expressed LAMP via viral infection to that of endogenous LAMP protein.

2) line 186-187, “Along the axon, antero- and retrograde transport of immature and mature lysosomes were observed”. How are mature lysosomes distinguished from immature ones? In reference 44, Li et al. concluded that “mature” lysosomes are restricted from axons.

3) Fig. 1a legend, a more detailed description of the RUSH system would be useful to readers who are not familiar with the system.

Reviewer #3

(Remarks to the Author)

In this manuscript (NCOMMS-24-24592-T), Li et al. investigate the delivery of newly synthesized endo-lysosomal membrane proteins LAMP1/2 to lysosomes in neurons. By combining selective retention and proximity labeling proteomics, the authors develop a system, which they refer to as POTATOMap, to define the interactome of LAMPs as they traverse the biosynthetic and endolysosomal pathways in neurons. The authors find distinct proteins enriched at specific time points corresponding to biosynthetic LAMP trafficking from the ER to Golgi to endo-lysosomes, identifying transport and fusion machineries required for LAMP targeting to neuronal lysosomes. Using live-neuron imaging, the authors validate and extend these findings to axons, revealing a role for biosynthetic LAMPs in replenishing axonal lysosomes, delivering newly synthesized synaptic proteins, and RNA granule hitchhiking. Based on these findings, the authors suggest that POTATOMap can be used to define the interactome of biosynthetic protein trafficking from origin to destination in health and disease.

Overall, this is a well-executed study that leverages new technologies to address how biosynthetic LAMPs are sorted and delivered to lysosomes in neurons and the nature of these compartments in axons. As the authors point out, this is an understudied area but an important question, given that several neurological disorders are linked to defects in lysosome biogenesis and function. The data are of high quality, and the conclusions are largely supported by the data. However, there are several points and concerns that require additional data and clarification, as detailed below.

1. The authors show that biosynthetic LAMPs are sorted into compartment(s) distinct from pre-existing lysosomes at 1 hour and reach lysosomes in the cell body within 4 hours. This seems to also be the case for axons, but how is it that biosynthetic LAMPs display similar trafficking kinetics (time points) in the cell body vs distal regions given the extended long morphology

of axons? Are all imaged axon segments in proximal regions? Based on the kymographs, biosynthetic LAMPs seem to transport at relatively high rates in axons. Can the velocity of these compartments be measured? In addition, can the co-distribution of RUSH-LAMP1 with STX6, Rab11, Rab7, and LAMTOR4 be validated in axons as demonstrated for the cell body? A more detailed characterization and validation of the system in delivering lysosomal membrane proteins to lysosomes in axons would strengthen the neuronal domain-specific aspects of the study.

2. The POTATOMap proteomics data did not reveal any lysosomal hydrolase enzymes that co-traffic with biosynthetic LAMP cargos. These contents are key for lysosomal degradation capacity. Does the current dataset indicate different cargos carrying membrane proteins and lysosomal luminal contents for their biosynthetic trafficking pathways in neurons? Related to this, does the authors' data and model shed light on whether newly synthesized LAMPs are delivered to lysosomes via direct and/or indirect targeting in neurons? Can POTATOMap be adapted to study lysosomal hydrolase trafficking to lysosomes via M6P-dependent and independent routes distinct from lysosomal membrane protein trafficking?

3. The authors suggest that POTATOMap can be used to identify the neuronal domain-specific trafficking machineries of biosynthetic LAMPs. However, the authors go on to study neuronal cell bodies and axons but not dendrites. While it is understood that proteomics data showed enrichment of processes associated with the axonal domain, can POTATOMap be used to map biosynthetic LAMP compartments and machineries in dendrites? Is RUSH-LAMP trafficking detected in dendrites, or is biosynthetic LAMP sorting and trafficking specific to axons? This would be of interest given the specificity of some LAMP-motor complexes for driving transport specifically into axons vs dendrites.

4. The POTATOMap dataset hits include ARL8A, ALR8B, KIF1A, and KLC2; all of these motors and adaptors drive axonal anterograde transport. Have the authors found any retrograde motors/adaptors in the dataset? If not, does it suggest that biosynthetic LAMP cargos specifically recruit motors moving towards the periphery (non-neuronal cells) and distal compartments (neurons)?

5. In Figure 5, the authors show the dependence of biosynthetic LAMP transport into axons on ARL8 and kinesins 1 and 3. Interestingly, Figure 5a shows that BORC subunits are also significantly increased between 20 minutes and 1 hour. Does biosynthetic LAMP transport depend on BORC?

6. In Figure 7b, some of the events denoted by arrows could represent passing RNA granules rather than hitchhiking; a kymograph showing co-trafficking of RNA granule proteins with RUSH-LAMP1 one hour after release would provide stronger support that RNA granules hitchhike on biosynthetic LAMP compartments for their axonal delivery. To further demonstrate RNA granule transport on biosynthetic LAMPs, can the authors show co-transport of RNA granules with RAB6A (a marker for biosynthetic LAMP1 compartments in nature, as the authors suggested) in axons? Because RNA granules were previously shown to hitchhike on axonal lysosomes, a more detailed characterization of RNA granule co-transport with biosynthetic LAMPs would be important.

7. It is difficult to determine which images are from endogenous staining vs overexpression. As the authors point out, a possible limitation of POTATOMap is that proteins may spill over to other delivery routes upon overexpression. Can more detail be provided on how proteins are being visualized (GFP, affinity tag, endogenous staining, etc.) in each figure? In addition, while line scan analyses are helpful to emphasize a particular finding, measuring colocalization may offer a more robust quantification approach that yields more information.

8. The discussion would benefit from mentioning how POTATOMap compares to TransitID (PMID: 37385249), a recently published method from Alice Ting's lab used to map endogenous protein trafficking. Because the authors suggest that POTATOMap can be applied to biosynthetic protein trafficking in health and disease, the discussion should include a paragraph comparing the two systems.

9. The discussion would also benefit from mentioning the finding that around 15% of all LAMP1-positive organelles in axons were post-Golgi carriers in nature, thus supporting a previous report that simply relying on LAMP1/2 staining is not accurate to assess neuronal lysosome distribution, trafficking, and functionality under physiological and pathological conditions (Cheng et al., JCB 2018, PMID: 29695488)

Minor points:

In line 221, a citation is missing.

In line 244, should be "carriers" instead of "carries."

In line 294, should be "iPSC" instead of "IPSC."

In Figures 5b, 6f, 6g, an evaluation of knockdown efficiency should be presented.

Reviewer #4

(Remarks to the Author)

I co-reviewed this manuscript with one of the reviewers who provided the listed reports. This is part of the Nature Communications initiative to facilitate training in peer review and to provide appropriate recognition for Early Career

Researchers who co-review manuscripts.

Version 1:

Reviewer comments:

Reviewer #2

(Remarks to the Author)

The authors have satisfactorily addressed my previous comments.

Reviewer #3

(Remarks to the Author)

In the revised version of their manuscript (NCOMMS-24-24592A), Li et al. have addressed all of my concerns with new data and expanded discussions. The study is novel and innovative, offering important insights into the mechanisms of biosynthetic LAMP trafficking. It also serves as an excellent resource for understanding the biosynthetic interactome of lysosomal proteins in neurons. In my opinion, this study will be of interest to the lysosomal field, particularly to those focused on neurobiology, and I recommend its publication in Nature Communications.

Reviewer #4

(Remarks to the Author)

Point-by-point response to the reviewers:

Reviewer #1 (Remarks to the Author):

This manuscript described the dynamic biosynthetic LAMP trafficking and interactome in neurons by combining RUSH microscopy and APEX proteomics strategies, namely, POTATOMap. Using this method, authors were able to track the movement of LAMP through the ER, Golgi and lysosome over different time points. Interestingly, this method found that immature and mature lysosomes enter the axon as separate compartments with distinct protein features, and that RNA granules traffic along the axons by attaching to biosynthetic LAMP compartments.

Overall, this is an elegant study that introduced a new methodology and revealed important roles of axonal lysosomes in delivery newly synthesized axonal synaptic proteins via kinesin and adaptor proteins and interactions with RNA granules. But the unique features/advantages of POTATOMap are not clearly discussed, and how are the proteomics results directly compared with the existing Lyso-IP or Lyso-APEX approaches are not provided. The authors also need to be careful to avoid overdrawing conclusions from the microscopy data. My main comments are listed below:

R: We thank the reviewer for his/her positive assessment of our study and the constructive input on our manuscript. See below point-by-point response to the comments.

Major Comments:

- Since the RUSH system is frequently used in this manuscript, including a brief description of how RUSH work would be helpful in the introduction section.

R: We thank the reviewer for raising this point. We have added two brief descriptions of how the RUSH system works in the Introduction and in the result section related to Fig. 1a; general and more specific, respectively. Additions are marked in blue in the revised manuscript.

The RUSH system allows the visualization of the sorting of biosynthetic proteins to their final destinations by synchronizing protein release from the ER. This works through ER retention by heterodimerization of a Streptavidin (Strep) fused to a retention signal or 'hook' (KDEL) and Streptavidin Binding Peptide (SBP) fused to a protein of interest. Upon addition of biotin, which competes for binding to Strep, the protein is released from the hook, and trafficking is synchronized.

- Why combining RUSH and proximity labeling provide unique advantages is not clear.

R: We thank the reviewer for raising this point. We have added in the discussion the advantages of combining RUSH and proximity labeling.

Our POTATOMap provides the unique advantage to study cargo sorting and interactome from the biosynthetic pathway to destination, by combining the RUSH system with APEX2. The RUSH system allows for spatial control of synchronously secreted cargo from the ER to its desired destination. This has been only used for imaging until now, in which co-distribution with known proteins is required, as mentioned in Introduction. On the other hand, the use of proximity labeling of lysosomal proteins at the steady state or lyso-IP has provided novel insights about lysosomes. Our approach is unique as it allows to create temporal snapshots of the interactome of LAMP at deliberately chosen times and locations from the biosynthetic pathway to the endolysosomal system. Our approach has allowed us to reveal potential key transient interactors, for an unbiased identification of key players not only regulating protein trafficking to lysosomes, but also novel roles of newly secreted biosynthetic compartments.

•In line 155, the authors mentioned several overlapping proteins from POTATOMap with previously published lyso-IP and lyso-proximity labeling proteomics. It would be helpful to provide a direct comparison of the proteome coverage (for example venn diagrams) among POTATOMap, Lyso-IP, and Lyso-APEX proteomics.

R: We thank the reviewer for this comment. In the original manuscript we mentioned that POTATOMap at 4h identifies most lysosomal proteins previously identified in lysosome proteomic datasets. We have now added a new **Supplementary Fig. 3** with venn diagrams comparing POTATOMap 4h to different lysosome proteomics approaches (see figure below): LAMP1-APEX2 dataset from Liao et al., 2019 in iPSC derived neurons (**a**); endogenous knock-in LAMP1-APEX2 dataset from Frankenfield et al., 2020 (**b**); TagLyso-IP in neuro-2a cells from Krogsaeter et al., 2023 (**c**) and TagLyso-IP performed in mouse from Laqtom et al., 2022 (**d**). Only significantly changing proteins ($D > 1, p < 0.05$) from the POTATOMap dataset at 4H were used for consistency. We understand the limitation of comparing different techniques and cellular models, however we do find key lysosomal proteins (overlapping regions) shared between datasets (associated with lysosomal membrane such as trafficking machineries and vesicle fusion complexes), as well as other known lysosomal proteins that are not shared between POTATOMap and other techniques, as shown in GO terms. For lyso-IP (**d**), we do not detect luminal lysosomal proteins as our APEX is present in the cytosolic domain aimed to identify potential regulators. A few shared proteins, as well as a few unique proteins from the different studies are highlighted. The full list comparing proteomics and GO analysis is included in **Source Data 1**.

•The authors also missed several key citations from recent years that used LAMP1-proximity labeling to study lysosome interactome. They should be cited in this manuscript:

Bhattacharya et al., Nat Cell Biol, 2023 (<https://pubmed.ncbi.nlm.nih.gov/37024685/>);

Hasan et al., Mol Neu. 2023 (<https://pubmed.ncbi.nlm.nih.gov/37974165/>);

Tan et al, Nature, 2022(<https://pubmed.ncbi.nlm.nih.gov/36071159/>)

R: We thank the reviewer for suggesting these important papers using LAMP1-proximity labeling, we have added citations to these papers in the manuscript.

•In Figure 2a, the authors show that POTATOMap was conducted using a 20 min, 1 hr, and 4 hr timepoint. What's the rationale of selecting these time points? What would happen after 4 hours?

R: We defined these timepoints through an optimization process to reflect different locations of LAMP from the biosynthetic pathway to the endolysosomal system. 20 minutes was used as a baseline timepoint for ER and Golgi residence since based on our imaging data, biosynthetic LAMP does not exit as post-Golgi carriers until 30 minutes after biotin addition. 1 hour was designated as the Golgi and post-Golgi timepoint where the majority of the biosynthetic LAMP is exiting from the trans-Golgi network in carriers that are not lysosomal in nature and lack hydrolase activity (Figs. 1 & 4). 4 hours was chosen because the majority of biosynthetic LAMP compartments at this timepoint colocalize with existing LAMP1-labelled lysosomes and harbour cathepsin D (hydrolase) activity labelled with SirLyso (Figs. 1 & 4). We have explained in the results section related to Fig. 2 that these timepoints were determined based on our imaging data in Fig. 1. We believe that beyond this 4-hour timepoint it would be similar to expressing a normal LAMP-GFP or LAMP-APEX2, as we obtain most endolysosomal proteins in our proteomics at 4h when comparing with normal LAMP-APEX (**Supplementary Fig. 3**). We have now also included images of RUSH-LAMP 24-hour post-biotin release in **Supplementary Fig. 1b & c**. See also below.

•In Figure 5, line 238, the authors estimated around 15% of LAMP1-positive organelles correspond to biosynthetic LAMP compartments. Similarly in Figure 7, line 307-309, the authors estimated the percentage of LAMP compartments. How are these percentage values calculated and quantified? Do they have biological replicates to support the reproducibility?

R: In our original experiments in Fig. 5 we quantified 3 individual biological replicates, with a relatively low number of cells analyzed. We have therefore included a 4th biological replicate to further strengthen our quantification and observation (Fig. 5I). To estimate the amount of biosynthetic LAMP compartments, present in the axon, we first counted the number of GFP-RAB6 and LAMP1-RFP in separate channels, then the two channels were merged and puncta that showed overlap of at least 90% of all pixels (which is in this case at least 4-8 pixels depending on vesicle size) would be considered colocalization.

•Overexpressing LAMPs have been previously shown to cause mislocalization to the plasma membrane. Have the authors observed this possible problem?

R: We thank the reviewer for raising this point. We are aware of LAMP over-expression artifacts, and we have tested and titrated different expression levels of RUSH-LAMPs (from 250ng DNA to 800ng DNA) in neurons, and we opt for cells that expresses the construct at relatively low levels in live imaging experiments. In short, we were not able to observe plasma membrane localization in neurons expressing low level of RUSH-LAMP plasmid with live imaging or immunofluorescence staining, and we recommend using low amounts of DNA (250ng RUSH-LAMP plasmid per 100k cells in 12-well plates). From our POTATOMap gene ontology analysis across all timepoints, we did not observe evident plasma membrane proteome being captured, which is in line with our live and fixed cell imaging where we did not observe distinct plasma membrane localization.

Minor comments

•Please make sure all abbreviations are spelled out in the manuscript. For example, TGN has been used multiple times in the manuscript, does it mean trans-Golgi network?

R: We thank the reviewer for the feedback. We have adjusted the necessary abbreviations in the manuscript and spelled out the abbreviations when it was first used.

•Please carefully read through the entire manuscript to avoid typos and inconsistency of the use of words, symbols, and capitalization.

R: We thank the reviewer for the feedback. We have proofread the manuscript for typos and inconsistency.

Reviewer #2 (Remarks to the Author):

In this study, the authors developed a novel system they dubbed POTATOMap. The system combines the “retention using selective hook (RUSH)” to synchronize the “releasing” of proteins from the ER and the APEX-based proximity labeling-based proteomics to reveal the interactomes at various time points after protein releasing. By applying the system to the lysosome-related proteins LAMPs in cultured neurons, the authors “released” LAMP proteins and monitored their interactomes at various time points as they are delivered to lysosomes. The studies reveal the gradual changes of the interactome environment and provide a snapshot of the system with an unprecedented temporal resolution. Several hundred proteins are discovered that are at some point within the vicinity of the LAMPs.

There are some novel interesting findings in the study. For example, the authors discovered that at ~1 hr after the “release”, the LAMP interactome includes some of the usual suspects such as those involved in Golgi- and endosome-transport, but also proteins involved in RNA processing and synaptic functions, before lysosome “functional” proteins such as mTOR were found ~4 hr after the “release”. Using live cell imaging and shRNA knock-down, the authors also demonstrated the significance of some of the components they found in the interactomes in functions such as organelle transport.

The studies described in the article are comprehensive and multi-disciplined. Many of the findings are novel and reveal new lysosome biology. The studies will be useful for researchers in the field of lysosome biology. Perhaps more importantly, some of the “-omics” findings in the studies will be the basis for future detailed mechanistic studies.

The studies are well-designed. The presentation is clear and easy to understand. I have only 3 minor comments.

R: We thank the reviewer for his/her positive assessment of our study and the constructive input on our manuscript.

1) The authors claim that they “map the endogenous interactome”. The interactomes, are, however, from that of exogenously expressed LAMP that is fused with other proteins (RUSH-LAMP1-V5-APEX2). It should be noted that the interactomes are not necessarily the same as those of the endogenous LAMP, especially if the protein is over-expressed. At minimum, it’d be useful if experiments were performed to compare the protein level of the exogenously expressed LAMP via viral infection to that of endogenous LAMP protein.

R: We thank the reviewer for raising this important point. As we have mentioned to Reviewer #1, we use a low amount of DNA for transfection for imaging data (250ng RUSH-LAMP plasmid per 100k cells in 12-well plates). With this DNA concentration we do not observe the reported typical artifact of mislocalization of LAMP to the plasma membrane. For proteomics, lentiviral expression showed even lower LAMP expression than for transfection. From our POTATOMap gene ontology analysis across all timepoints, we did not observe evident plasma membrane proteome being captured, which is in line with our live and fixed cell imaging where we did not observe distinct plasma membrane localization. We have tried to perform western blot experiments to compare protein levels of exogenously expressed LAMP via lentiviral infection in rat neurons; unfortunately, the monoclonal LAMP1 antibody (LY1C6), didn’t work in our hands. We therefore cannot determine the precise amount of exogenous LAMP1 expression. However, we agree that exogenously expressed LAMP does not strictly reflect the endogenous interactome, we have hence corrected our statement to ‘map the interactome’.

2) line 186-187, “Along the axon, antero- and retrograde transport of immature and mature lysosomes were observed”. How are mature lysosomes distinguished from immature ones? In reference 44, Li et al. concluded that “mature” lysosomes are restricted from axons.

R: We thank the reviewer for raising this point. We used SirLyso to probe for cathepsin D activity in lysosomes to distinguish mature lysosomes from their immature counterparts. We are aware of the limitations of probing cathepsin D activity as they may be undergoing acidification to mature. We observe that the majority of LAMP1-positive SirLyso-positive compartments (mature lysosomes) are retrogradely transported along the axon, while LAMP1-positive SirLyso-negative immature lysosomes can be found trafficking in anterograde and retrograde directions along the axon. Occasionally, as seen in Fig. 4d-f, LAMP1-positive and SirLyso-positive compartments can be seen undergoing anterograde trafficking. The presence of both immature (majority) and mature (minority) lysosomes are consistent with previous papers such as Farias et al., PNAS 2017; Xiu-Tang Cheng, JCB., 2018; Farfel-Becker et al., Cell Rep 2019.

3) Fig. 1a legend, a more detailed description of the RUSH system would be useful to readers who are not familiar with the system.

R: We thank the reviewer for raising this point. As mentioned earlier to Reviewer #1, we have added two brief descriptions of how the RUSH system works in Introduction and in the result section related to Fig. 1a; general and more specific, respectively. Additions are marked in blue in the revised manuscript.

The RUSH system allows the visualization of the sorting of biosynthetic proteins to their final destinations by synchronizing protein release from the ER. This works through ER retention by heterodimerization of a Streptavidin (Strep) fused to a retention signal or ‘hook’ (KDEL) and Streptavidin Binding Peptide (SBP) fused to a protein of interest. Upon addition of biotin, which competes for binding to Strep, the protein is released from the hook, and trafficking is synchronized.

Reviewer #3 (Remarks to the Author):

In this manuscript (NCOMMS-24-24592-T), Li et al. investigate the delivery of newly synthesized endo-lysosomal membrane proteins LAMP1/2 to lysosomes in neurons. By combining selective retention and proximity labeling proteomics, the authors develop a system, which they refer to as POTATOMap, to define the interactome of LAMPs as they traverse the biosynthetic and endolysosomal pathways in neurons. The authors find distinct proteins enriched at specific time points corresponding to biosynthetic LAMP trafficking from the ER to Golgi to endo-lysosomes, identifying transport and fusion machineries required for LAMP targeting to neuronal lysosomes. Using live-neuron imaging, the authors validate and extend these findings to axons, revealing a role for biosynthetic LAMPs in replenishing axonal lysosomes, delivering newly synthesized synaptic proteins, and RNA granule hitchhiking. Based on these findings, the authors suggest that POTATOMap can be used to define the interactome of biosynthetic protein trafficking from origin to destination in health and disease.

Overall, this is a well-executed study that leverages new technologies to address how biosynthetic LAMPs are sorted and delivered to lysosomes in neurons and the nature of these compartments in axons. As the authors point out, this is an understudied area but an important question, given that several neurological disorders are linked to defects in lysosome biogenesis and function. The data are of high quality, and the conclusions are

largely supported by the data. However, there are several points and concerns that require additional data and clarification, as detailed below.

R: We appreciate the reviewer for her/his positive comments and constructive feedback on our manuscript. We have added new experiments based on this reviewer's comments, which we believe further strengthens our conclusions.

1. The authors show that biosynthetic LAMPs are sorted into compartment(s) distinct from pre-existing lysosomes at 1 hour and reach lysosomes in the cell body within 4 hours. This seems to also be the case for axons, but how is it that biosynthetic LAMPs display similar trafficking kinetics (time points) in the cell body vs distal regions given the extended long morphology of axons? Are all imaged axon segments in proximal regions? Based on the kymographs, biosynthetic LAMPs seem to transport at relatively high rates in axons. Can the velocity of these compartments be measured? In addition, can the co-distribution of RUSH-LAMP1 with STX6, Rab11, Rab7, and LAMTOR4 be validated in axons as demonstrated for the cell body? A more detailed characterization and validation of the system in delivering lysosomal membrane proteins to lysosomes in axons would strengthen the neuronal domain-specific aspects of the study.

R: We thank the reviewer for the feedback. We have added an illustration of our imaging strategy in **Fig. 4a** (see also below). In brief, all images were taken at the proximal axon approximately 50 microns away from the AIS.

R: Biosynthetic LAMP predominantly transports in anterograde direction at the proximal axon after 1 hour of biotin addition, while more bidirectional and stationary carriers can be observed further into the axon. We have now quantified the velocities of RUSHLAMP2a at 1h (anterograde: $1.51 \pm 0.89 \mu\text{m/s}$; retrograde: $0.83 \pm 0.59 \mu\text{m/s}$) and at 4h (anterograde: $1.06 \pm 0.77 \mu\text{m/s}$; retrograde: $0.75 \pm 0.41 \mu\text{m/s}$), included in **Supplementary Fig. 5g** (see also below). The quantified velocities potentially suggest a switch from fast motor transport with KIF1A after the AIS at 1h to slower transport with KIF5A/B/C at 4h, while retrograde transport with dynein did not display any speed differences. We have also added a discussion regarding velocity and our KD experiments for motors at 1h, in which both kinesin-1 and kinesin-3 are required for the transport of biosynthetic LAMP.

R: We have added new quantifications for the co-distribution of RUSH-LAMP1 with endogenous STX6, GFP-RAB11, GFP-RAB7 and endogenous LAMTOR4 in cell body and axon, included in **Supplementary Fig. 4a-d** (see also below). Both STX6 and GFP-RAB11 in the cell body showed reduced colocalization from 1 h to 4 h, while colocalization of both markers in the axon with RUSH-LAMP1 is minimal at 1h and 4h; GFP-RAB7 and LAMTOR4 showed an increased colocalization from 1 h to 4 h in both soma and axon, agreeing with our proteomics and live imaging data.

2. The POTATOMap proteomics data did not reveal any lysosomal hydrolase enzymes that co-traffic with biosynthetic LAMP cargos. These contents are key for lysosomal degradation capacity. Does the current dataset indicate different cargos carrying membrane proteins and lysosomal luminal contents for their biosynthetic trafficking pathways in neurons? Related to this, does the authors' data and model shed light on whether newly synthesized LAMPs are delivered to lysosomes via direct and/or indirect targeting in neurons? Can POTATOMap be adapted to study lysosomal hydrolase trafficking to lysosomes via M6P-dependent and independent routes distinct from lysosomal membrane protein trafficking?

R: We thank the reviewer for raising this interesting observation. We reasoned that because the APEX2 module is tagged at the C-terminal short cytosolic tail of LAMP1, luminal hydrolases such as cathepsins that predominantly resides in the lumen of membrane compartments are likely not biotinylated and can therefore not confidently be detected. This can be seen in the venn-diagram comparing TagLysoIP from Laqtom et al., 2022 with our POTATOMap 4 h timepoint in the new **Supplementary Fig. 3d**, where a lot of luminal hydrolases and modifying enzymes are missed. Our current POTATOMap dataset therefore cannot exclude the fact that luminal hydrolases might also be present after their segregated exit from the TGN (**Supplementary Fig. 7a**), but further investigations and more detailed characterization is needed. Our colocalization analysis between STX6, RAB11, RAB7, RAB6 and LAMTOR4 in the axon, and the lack of plasma membrane localization of biosynthetic LAMP along the axon would suggest there could be a direct targeting to lysosomes. Although we cannot rule out other routes could also take place in different neuronal domains, such as in the soma and dendrite where we observe biosynthetic LAMP compartments in close proximity to RAB11 recycling endosomes (see also response to comment #3). These observations would suggest biosynthetic LAMP can be targeted to different destinations depending on the neuronal subdomain, and our study offers new directions for further investigations. POTATOMap can be adapted to study other lysosomal protein trafficking, for instance a RUSH version of M6PR tagged with APEX2 at the cytosolic side would be an interesting candidate to study M6PR trafficking and retrieval in non-polarized and polarized cells. We recommend using transmembrane proteins as a target because it is only feasible to detect trafficking machineries and other machineries of interest in the cytosol.

3. The authors suggest that POTATOMap can be used to identify the neuronal domain-specific trafficking machineries of biosynthetic LAMPs. However, the authors go on to study neuronal cell bodies and axons but not dendrites. While it is understood that proteomics data showed enrichment of processes associated with the axonal domain, can POTATOMap be used to map biosynthetic LAMP compartments and machineries in dendrites? Is RUSH-LAMP trafficking detected in dendrites, or is biosynthetic LAMP sorting and trafficking specific to axons? This would be of interest given the specificity of some LAMP-motor complexes for driving transport specifically into axons vs dendrites.

R: We thank the reviewer for raising this point. Our POTATOMap dataset also identified some dendritic proteins such as PSD95 and NLGN3. In our GO cellular compartment analysis 'dendrite' is also present in both 1h and 4h (see source data 1). However, because of the abundance of axon-associated proteins, we decided to primarily focus on exploring the role of biosynthetic LAMPs in axonal processes. It is important to note that biosynthetic LAMP is also detected in dendrites in live and fixed neuron imaging as shown in new **Supplementary Fig. 4a-d** and would be of significant interest to further investigate the role and function of biosynthetic LAMP being targeted into dendrites.

4. The POTATOMap dataset hits include ARL8A, ALR8B, KIF1A, and KLC2; all of these motors and adaptors drive axonal anterograde transport. Have the authors found any retrograde motors/adaptors in the dataset? If not, does it suggest that biosynthetic LAMP cargos specifically recruit motors moving towards the periphery (non-neuronal cells) and distal compartments (neurons)?

R: We have included some of the dynein subunits detected in our revised **Fig. 5A** (see also below). DYNC1H1, DYNC1LI1 and DYNC1LI2 are detected with significance at 4 hours after biotin but not after 1 hour compared to motors and adaptors such as KIF1A and KLC2. It can be reasoned that biosynthetic LAMP predominantly recruits motors moving towards the periphery/distal compartments of neurons and only a minor population of vesicles contain dynein, which largely coincides with our live imaging data showing biosynthetic LAMP1 at 1 hour is predominantly undergoing anterograde trafficking to axons.

5. In Figure 5, the authors show the dependence of biosynthetic LAMP transport into axons on ARL8 and kinesins 1 and 3. Interestingly, Figure 5a shows that BORC subunits are also significantly increased between 20 minutes and 1 hour. Does biosynthetic LAMP transport depend on BORC?

R: We thank the reviewer for raising this interesting point. We have added data about the effects of BORCS5 shRNA on RUSH-LAMP1 transport into the axon in **Supplementary Figs. 6g-h** (see also below) and discussed in the result section related to biosynthetic LAMP transport. Knockdown of the BORC complex subunit BORCS5 caused only a slight reduction in the axonal trafficking of RUSH-LAMP1 at 1h, which aligned with our proteomics data, in which the BORC subunits SNAPIN and BLOC1S1 were only found associated to lysosomes at 4h.

We have also validated our BORCS5 shRNAs with quantitative RT-PCR in **Supplementary Fig. 6f** (see also below).

6. In Figure 7b, some of the events denoted by arrows could represent passing RNA granules rather than hitchhiking; a kymograph showing co-trafficking of RNA granule proteins with RUSH-LAMP1 one hour after release would provide stronger support that RNA granules hitchhike on biosynthetic LAMP compartments for their axonal delivery. To further demonstrate RNA granule transport on biosynthetic LAMPs, can the authors show co-transport of RNA granules with RAB6A (a marker for biosynthetic LAMP1 compartments in nature, as the authors suggested) in axons? Because RNA granules were previously shown to hitchhike on axonal lysosomes, a more detailed characterization of RNA granule co-transport with biosynthetic LAMPs would be important.

R: We thank the reviewer for raising this point. We have added imaging data in fixed cells quantifying the colocalization of RAB6A with RNA granule protein FXR1 in **Figs. 7 d-e** (see also below). We observe approximately 8% of RAB6A would be in close proximity and colocalize with FXR1, which coincides with our data estimating that at the steady state the biosynthetic pool of LAMP1 is about 11% (Fig. 5i).

We have added new observations in **Fig. 7h** (see also below) where after 1 h of biotin in the proximal axon, FXR1 RNA granules in close proximity to biosynthetic LAMP undergo fission.

We have also added still images and a kymograph indicating that FXR1 RNA granule can co-transport with biosynthetic LAMP in the proximal axon after 1 h of biotin in new Figs 7j-k.

All together we believe that our new observation could help strengthen our data suggesting biosynthetic compartments could potentially be another candidate to transport RNA granules into the axon. However, we would like to clarify an important observation. Whilst we occasionally observe RNA granules undergoing active transport in our rat hippocampal neuron culture at DIV5-7, we predominantly observe stationary RNA granules (both FXR1 and mRNA actin) in close proximity with stationary biosynthetic LAMP compartments at 1 h in proximal axon. This observation is consistent with previous studies such as Cioni et al., Cell 2019, in which they report that approximately 66-72% of RNA granules were stationary; and Wong et al., Neuron 2017 in which they report 40% of RNA granules were stationary.

7. It is difficult to determine which images are from endogenous staining vs overexpression. As the authors point out, a possible limitation of POTATOMap is that proteins may spill over to other delivery routes upon overexpression. Can more detail be provided on how proteins are being visualized (GFP, affinity tag, endogenous staining, etc.) in each figure? In addition, while line scan analyses are helpful to emphasize a particular finding, measuring colocalization may offer a more robust quantification approach that yields more information.

R: We thank the reviewer for the feedback. We have clarified in our figures with annotations indicating endogenous staining. Because the limited space in Figures, we did not add the tag for expressed proteins there. However, we have revised our figure legends to provide more clarity to our readers. We also agree that line scan is not quantitative, we have therefore included colocalization analysis of endogenous STX6, endogenous LAMTOR4, GFP-RAB11 and GFP-RAB7 in **Supplementary Fig. 4a-d** as mentioned in our response to reviewer #3 question 1.

8. The discussion would benefit from mentioning how POTATOMap compares to TransitID (PMID: 37385249), a recently published method from Alice Ting's lab used to map endogenous protein trafficking. Because the authors suggest that POTATOMap can be applied to biosynthetic protein trafficking in health and disease, the discussion should include a paragraph comparing the two systems.

R: We thank the reviewer for the comment. Although TransitID and POTATOMap answer different questions, we have compared both tools in our discussion.

Like the newly developed TransitID approach, POTATOMap enables time-resolved proteomics to study protein origin and destination. As TransitID is organelle-based proteomics, it could be used to identify which proteins (cargoes) are trafficked from one organelle to another. However, it cannot be used to study the trafficking pathway and trafficking machinery of these cargoes. POTATOMap is protein cargo-based proteomics, offering the opportunity to identify intermediate compartments and characterize novel trafficking machineries.

9. The discussion would also benefit from mentioning the finding that around 15% of all LAMP1-positive organelles in axons were post-Golgi carriers in nature, thus supporting a previous report that simply relying on LAMP1/2 staining is not accurate to assess neuronal lysosome distribution, trafficking, and functionality under physiological and pathological conditions (Cheng et al., JCB 2018, PMID: 29695488)

R: We thank the reviewer for raising this important point. We have added in our discussion that our data estimating the pool of biosynthetic LAMP is approximately 11% as shown in new Fig. 5i (See also below; additional 4th experiment performed), agreeing with the notion of a highly heterogenous population of LAMP1 positive compartments in axon, which are not mature lysosomes (Cheng et al. JCB 2018; reference added to the Discussion).

Minor points:

In line 221, a citation is missing.

R: We have added the necessary citation, now line 234-235.

In line 244, should be "carriers" instead of "carries."

R: We have made the necessary change.

In line 294, should be "iPSC" instead of "IPSC."

R: We have made the necessary change.

In Figures 5b, 6f, 6g, an evaluation of knockdown efficiency should be presented.

R: We thank the reviewer for raising this point. We have added in new **Supplementary Figs. 6a-d** (see also below) immunofluorescence data quantifying knockdown efficiency of shKIF5A/B/C and shARL8B with endogenous antibodies; **Supplementary Figs. 6e-f** (See also below) quantitative RT-PCR for shKIF1A and shBORCS5; and immunofluorescence data quantifying knockdown efficiency of shVAMP4 in new **Supplementary Fig. 7c** (See also below).

Reviewer #4 (Remarks to the Author):

R: We thank the reviewer for your valuable time for reviewing our manuscript and providing training to early career researchers.